# Network-Based Prediction of Side Effects of Repurposed Antihypertensive Sartans against COVID-19 via Proteome and Drug-Target Interactomes

**DOI:** 10.3390/proteomes11020021

**Published:** 2023-06-08

**Authors:** Despoina P. Kiouri, Charalampos Ntallis, Konstantinos Kelaidonis, Massimiliano Peana, Sotirios Tsiodras, Thomas Mavromoustakos, Alessandro Giuliani, Harry Ridgway, Graham J. Moore, John M. Matsoukas, Christos T. Chasapis

**Affiliations:** 1Institute of Chemical Biology, National Hellenic Research Foundation, 11635 Athens, Greece; despoina.kiouri.99@gmail.com (D.P.K.); xntallis@gmail.com (C.N.); 2Department of Chemistry, Laboratory of Organic Chemistry, National Kapodistrian University of Athens, 15772 Athens, Greece; tmavrom@chem.uoa.gr; 3NewDrug PC, Patras Science Park, 26504 Patras, Greece; k.kelaidonis@gmail.com; 4Department of Chemical, Physical, Mathematical and Natural Sciences, University of Sassari, Via Vienna 2, 07100 Sassari, Italy; peana@uniss.it; 54th Department of Internal Medicine, School of Medicine, National and Kapodistrian University of Athens, 11527 Athens, Greece; sotirios.tsiodras@gmail.com; 6Environment and Health Department, Istituto Superiore di Sanità, 00161 Rome, Italy; alessandro.giuliani@iss.it; 7Institute for Sustainable Industries and Liveable Cities, Victoria University, Melbourne, VIC 8001, Australia; 8AquaMem Consultants, Rodeo, NM 88056, USA; 9Pepmetics Inc., 772 Murphy Place, Victoria, BC V6Y 3H4, Canada; mooregj@shaw.ca; 10Department of Physiology and Pharmacology, Cumming School of Medicine, University of Calgary, Calgary, AB T2N 1N4, Canada; 11Institute for Health and Sport, Victoria University, Melbourne, VIC 3030, Australia; 12Department of Chemistry, University of Patras, 26504 Patras, Greece

**Keywords:** angiotensin receptor blockers, Sartans, coronavirus disease 19, angiotensin-converting enzyme 2, protein–protein interaction networks, drug–drug interaction prediction, off-target interaction prediction, gene ontology

## Abstract

The potential of targeting the Renin-Angiotensin-Aldosterone System (RAAS) as a treatment for the coronavirus disease 2019 (COVID-19) is currently under investigation. One way to combat this disease involves the repurposing of angiotensin receptor blockers (ARBs), which are antihypertensive drugs, because they bind to angiotensin-converting enzyme 2 (ACE2), which in turn interacts with the severe acute respiratory syndrome coronavirus 2 (SARS-CoV-2) spike protein. However, there has been no in silico analysis of the potential toxicity risks associated with the use of these drugs for the treatment of COVID-19. To address this, a network-based bioinformatics methodology was used to investigate the potential side effects of known Food and Drug Administration (FDA)-approved antihypertensive drugs, Sartans. This involved identifying the human proteins targeted by these drugs, their first neighbors, and any drugs that bind to them using publicly available experimentally supported data, and subsequently constructing proteomes and protein–drug interactomes. This methodology was also applied to Pfizer’s Paxlovid, an antiviral drug approved by the FDA for emergency use in mild-to-moderate COVID-19 treatment. The study compares the results for both drug categories and examines the potential for off-target effects, undesirable involvement in various biological processes and diseases, possible drug interactions, and the potential reduction in drug efficiency resulting from proteoform identification.

## 1. Introduction

The proper function of the human body is achieved by the interaction and perfect coordination of several systems and organs. RAAS is a crucial regulator of various body functions necessary for survival, such as blood volume and pressure regulation, sodium and water reabsorption, potassium secretion, and maintenance of the vascular tone [1,2]. Disturbance of this intricate hormonal regulatory system could potentially lead to acute or chronic conditions, such as various cardiovascular and renal disorders, including congestive heart failure (CHF), acute myocardial infarction (AMI), hypertension, and diabetic kidney disease (DKD) [1,3]. Until now, five main axes of the RAAS regulatory action have been reported, each of which utilizes different enzymes, receptors, key substrates, effector peptides, and/or leads to distinct downstream signaling pathways [4]. The five main axes are the classic angiotensinogen/renin/angiotensin-converting enzyme (ACE)/angiotensin II (ANG II)/angiotensin type 1 receptor (AT1R)/angiotensin I (ANG I)/ANG II/angiotensin-converting enzyme 2 (ACE2)/angiotensin 1–7 (ANG(1–7))/Mas receptor, the ANG II/alanine and proline-rich secreted protein (APA)/ANG III/angiotensin type 2 receptor (AT2R)/NO/cGMP, the ANG III/3-Arylpropiolonitriles (APN)/angiotensin IV (ANG IV)/insulin-regulated aminopeptidase (IRAP)/polyketide synthase (AT4) receptor, and the prorenin/renin/prorenin receptor (PRR or Atp6ap2)/MAP kinases ERK1/2/V-ATPase [4].

The dynamic equilibrium between the first two arms of the RAAS, ACE (the classical RAAS) and ACE2, is crucial for its regulation [5]. In the ACE pathway, ANG-I is formed as a result of the cleavage of angiotensinogen (AGT) (produced in the liver) from renin (produced in the kidney) [6]. Later on, ANG-I is cleaved by ACE into angiotensin II (ANG-II) (produced in the vascular tissue) that binds to two distinct protein receptors of the G protein-coupled family, angiotensin type 1 and type 2 (AT1R and AT2R, respectively) [7]. The AT1R pathway promotes vasoconstriction, cell growth, sodium, and water retention, as well as sympathetic activation. In contrast, the binding of ANG-II to the AT2R neutralizes the detrimental effects induced by the AT1R pathway [8]. Oppositely, ACE2 can produce angiotensin 1–9 (ANG 1–9) from ANG-I and angiotensin 1–7 (ANG 1–7) from ANG-II [6]. The production of ANG 1–7 limits the available ANG-II, thus reducing the activation of the AT1R while simultaneously binding to the MAS-receptor, resulting in vasodepressor, anti-inflammatory, anti-oxidative, and antiproliferative effects [5].

ACE inhibitors and ARBs (also named Sartans, Figure 1) are first-line drugs for hypertension but are also employed to treat certain cases of heart failure and chronic kidney disease [9]. On the one hand, ACE inhibitors impede the production of ANG-II, reducing the activity of both AT1R and AT2R [9]. However, because this enzyme is also a kininase, its inhibition raises the overall level of kinins. One of those, bradykinin, is correlated with a series of side effects such as cough, but on top of that, in rare cases, with inhibitor-induced angioedema, which is a potentially life-threatening emergency [9]. On the other hand, ARBs were engineered as a substitute for patients who could not endure the adverse events of ACE inhibitors. These drugs specifically block the AT1R while augmenting the activation of the AT2R [10]. Additionally, ARBs can limit inflammation along with endothelial and epithelial dysfunction in various organs. More specifically, there is clinical evidence suggesting that the integrity of the lung’s endothelial barrier, which might be disrupted in the case of a viral infection, can be protected by ARBs [11].

Recently, ARB repurposing efforts have been made for the treatment of COVID-19 in order to address the urgent need for effective therapy for the disease while minimizing the costs, time, and uncertainty that accompany most drug development strategies [12,13]. The rationale behind this work is the fact that Sartans, as AT1R antagonists, are substantially ANG-II mimic molecules and thereby are expected to bind to the ACE2 enzyme [12]. Previously, the development and in silico study of biSartans, a novel type of Sartans with two anionic biphenyl tetrazole moieties, were reported [12,13]. These molecules were found to bind stronger than Sartans not only to the AT1R but also to the Receptor Binding Domain (RBD)/ACE2 complex [13]. It is well known that the spike S-protein of SARS-CoV-2 binds to ACE2 through its RBD, thus initiating membrane fusion between the virion and the cell [14]. One of the antiviral drugs currently approved by the FDA for emergency use for the treatment of mild-to-moderate COVID-19 in certain adults and pediatric patients is Pfizer’s Paxlovid (Figure 1). Paxlovid is a combination of PF-07321332 (Nirmatrelvir), an inhibitor of the crucial for the viral proliferation of 3-chymotrypsin-like protease (3CL^pro^), and Ritonavir, which acts as a pharmacokinetic boosting agent and is also used for the treatment of the Human Immunodeficiency Virus (HIV) [15]. As this drug is a strong cytochrome P450 (CYP) 3A4 inhibitor, co-administration with drugs that are highly dependent on CYP3A4 for clearance or potent CYP3A4 inducers could potentially lead to elevated concentrations of those drugs and greatly reduce Paxlovid plasma concentrations, respectively [16]. The problem that arises is that many drugs used regularly by a large percentage of the population, such as statins, antiarrhythmics, and antipsychotics, are metabolized by CYP3A4, and thus there is a great chance of drug interaction between those drugs and Paxlovid that could conceivably lead to side effects [17,18]. Most of the ARBs (i.e., losartan, irbesartan, candesartan, azilsartan, and valsartan) are predominantly metabolized by another CYP isomorph, CYP2C9, and to a smaller extent by CYP3A4 [19,20,21]. In the case of telmisartan, eprosartan, and Olmesartan, however, the CYP system does not participate in their metabolism [21].

**Figure 1 proteomes-11-00021-f001:**
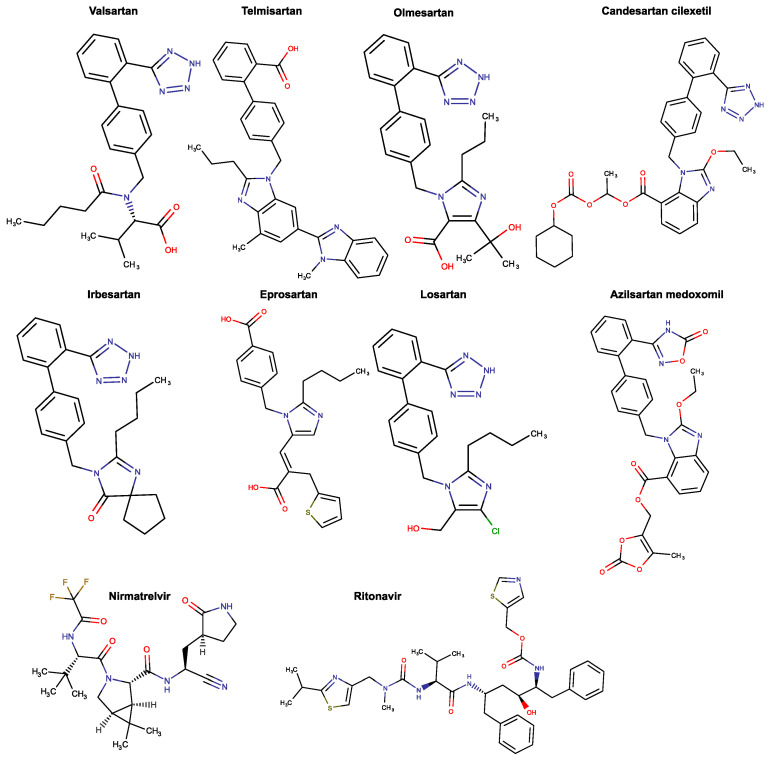
Chemical structures of Valsartan, Telmisartan, Olmesartan, Candesartan cilexetil, Irbesartan, Eprosartan, Losartan, Azilsartan medoxomil, Ritonavir, and Nirmatrelvir. Note that some of these may be deprotonated at physiological pH [13,22,23]. Illustrations were made with MarvinSketch, version 22.22.0 [24].

In this work, a network-based methodology was developed as an initial prediction of the potential for side effects resulting from protein off-target and/or drug–drug interactions that could occur when a drug is repurposed. It has been suggested that the proteins that directly interact with a drug’s protein target (first neighbors) could act as off-targets for the specific drug in question, other drugs that share the same drug target with the drug in question, and drugs that target the drug’s first neighbors in the network [25]. Additionally, the drugs that share a drug target or target neighbor proteins in the human interactome could also participate in drug–drug interactions [26]. Protein off-target and drug–drug interactions can in some cases be beneficial, for example, in drug repurposing or drug combinations for efficiency enhancement. However, in other cases, they can also lead to side effects. In this study, a network-based bioinformatics approach was used to explore the potential of FDA-approved antihypertensive drugs, Sartans, for repurposing against SARS-CoV-2. The analysis investigated their off-target interactions, effects on biological processes, and potential interactions with other drugs by integrating experimental and publicly available proteome interactome and drug-target data. As a comparison, the same methodology was also applied for Pfizer’s Paxlovid, as there is an evident structural similarity of its components, especially Ritonavir, with Sartans. This comparison could provide further insight into what would happen if Sartan was co-administered with Paxlovid. Some steps of the methodology were also applied to Perphenazine, a drug that has nothing to do with ACE2 or COVID-19, as a sort of “negative control test” for the relevance of the obtained results. Perphenazine is an antipsychotic drug that displays comparable effectiveness to Haloperidol and is used for the management of schizophrenia symptoms as well as the control of severe nausea and vomiting in adult patients [27,28].

## 2. Materials and Methods

The methodology workflow is presented in Figure 2. Το identify the specific proteins that are targeted by the drug in question, the DrugBank database was used [29]. DrugBank is a web-enabled database that includes thorough molecular details regarding drugs, their mechanisms of action, interactions, and targets [30,31,32,33,34]. To ascertain the structural similarity between the drugs in question, the Tanimoto index was calculated with the use of the Similarity Workbench of ChemMine Tools [35,36]. Additionally, DrugBank was also used to ascertain whether the human protein targets of the drug in question are also targets of other drugs approved at a certain moment in time in at least one jurisdiction. Next, using the Human Protein Atlas, which has all the human proteins mapped in cells, tissues, and organs through the integration of multi-omics data [37,38], the expression of each drug receptor was identified.

Human interactome data was downloaded from the Protein InteraCtion KnowLedgebasE (PICKLE) meta-database at the Universal Protein Resource (UniProt) level [39,40]. PICKLE web-source consists of the direct protein-protein interactome of the human proteome, integrating publicly available experimentally supported protein–protein interactions (PPIs) [41,42,43,44]. The protein targets for the drugs in question were projected into the human interactome, and binary PPIs between protein targets and their first neighbors (i.e., the proteins with which they physically interact) were extracted. The proteome interactomes and drug-target data were integrated to construct protein–drug interaction networks that include the drugs of interest (Sartans or Paxlovid) and their corresponding receptors, along with any other drugs that may also target these receptors. Additionally, the networks include the first neighbors of the receptors in the human interactome, as well as any drugs that target these first neighbors. The proteome interactomes and drug-target data were integrated for the construction of protein–drug interaction networks that include the drugs of interest (Sartans or Paxlovid) and their corresponding receptors, along with any other drugs that may also target these receptors. Additionally, the networks include the interactors of the receptors in the human interactome, as well as any drugs that target these interactors. The networks were constructed as described in previous works [45], including the import of mined PPI data from databases, visualization tools to adjust the layout, node size and color, edge thickness and color, and analysis tools using clustering algorithms for calculation of centrality measures in Cytoscape (version 3.9.1) [46]. After the network construction, the node degree, betweenness centrality, and closeness centrality calculations of the interactors of the drugs in question was also performed in Cytoscape. For the network comparison, the Jaccard index (a similarity coefficient equivalent to the Tanimoto index and correspondent to 1-dJ, where dJ is the well-known Jaccard distance for sparse matrixes [47]) was used to compute the similarity (a proxy of the probability of interaction) in terms of first neighbors between the protein–drug networks.

A plug-in of Cytoscape named Biological Networks Gene Ontology tool (BiNGO) was used for the functional annotation of the drug in question’s protein targets. BiNGO is an open-source Java tool for the identification of Gene Ontology (GO) keywords that are notably overrepresented in a given set of genes [48]. The statistical test implemented for the calculation of the overrepresentation is the hypergeometric test, whereas the False Discovery Rate (FDR) correction is carried out with the Benjamini–Hochberg method. BiNGO uses the term-for-term approach that detects overrepresentation of GO terms individually, and in our work, it is used as a validation of the pathways that the drug targets are involved in.

For the functional annotation of the interactors of the drugs’ targets, the Protein Analysis THrough Evolutionary Relationships (PANTHER) database was used [49]. PANTHER is a publicly available knowledge base that stores the outcomes of a lengthy phylogenetic reconstruction containing both computational and manual operations as well as quality control stages [50]. The statistical test used to determine the over-represented GO terms is the Fisher exact test. The Benjamini–Hochberg False Discovery Rate (FDR) is also computed, and all displayed GO terms have an FDR < 0.05 [51].

The gene–disease association of the protein targets of the drugs in question and their interactors was performed on the DisGeNET (version 7) platform, which contains an extensive catalog of genes and genomic variants associated with human diseases [52].

The proteoform identification of protein targets and their first neighbors was performed through the iteration of data mined from UniProt, the Online Mendelian Inheritance in Man (OMIM), and GeneCards. Proteoforms resulting from post-translational modifications (PTMs) were identified from UniProt, while those resulting from allelic variants of genes were extracted from OMIM, a comprehensive catalog of human genes, genetic conditions, and attributes that emphasizes the molecular linkage of genetic variation to phenotypic expression [53,54]. Additionally, proteoforms resulting from alternative splicing were extracted from the GeneCards database, which integrates information from multiple web sources about all annotated and predicted human genes [55,56].

## 3. Results

### 3.1. Retrievement of Drug–Protein Target Associations

There is an evident structural similarity between Sartans and Paxlovids that was verified by the calculation of the Tanimoto index. The Tanimoto index, along with the Dice index, cosine coefficient, and Soergel distance, have been proven to be the best (and, in some cases, equivalent) metrics for similarity computations [57]. In this case, the Tanimoto index of Sartans and Ritonavir ranges between 0.13 and 0.23 and the one of Sartans and Nirmatrelvir ranges between 0.04 and 0.23. Since the index is relatively low but different from zero (the total superposition scoring 1), the two kinds of drugs share some common structural motifs, and thus it is worth comparing them with one another to find out what would happen in a case of co-administration. The complete table of the Tanimoto index for each Sartan and the components of Paxlovid can be found in Appendix A.

Based on the DrugBank database, all Sartans bind to the AT1R. These receptors, as mentioned above, are vital effectors of the RAAS and are found in various cell types, as ANG II is a molecule with a wide range of actions [57]. Two of the approved Sartans (Telmisartan and Irbesartan) bind to one more protein each, which is also a target of many other drugs. Irbesartan binds to c-JUN, which is a transcription factor (TF) AP-1 subunit. c-JUN, as a basic leucine zipper (bZIP) TF, takes part in many different cell functions, including proliferation, apoptosis, survival, cancer, and tissue morphogenesis [58]. Proliferator-activated receptor γ (PPARγ) is one of the targets of Telmisartan. This protein belongs to the nuclear receptor superfamily of TFs and is a main regulator of the differentiation, maturation, and function of colon, breast, prostate, bladder, fat, and immune system cells [59]. If Sartans are repurposed for the treatment of COVID-19, they are going to target the ACE2 receptor. ACE2 is a homologue of ACE and mainly contributes to the regulation of ANG II levels but also hydrolyzes some proteins such as bradykinin [60,61]. This zinc metalloenzyme and monocarboxypeptidase is a Type 1 integral membrane glycoprotein [62] and is located both on the surface of endothelial and epithelial cells of the kidney, heart, uterus, placenta, and retina, as well as other tissues [60]. However, there is also a secreted form of this enzyme in the blood [60].

Paxlovid has two drug receptors, one for each of its active components. Nirmatrelvir, which is the antiviral component, binds to 3CL^pro^, whereas Ritonavir (Figure 1), which is essentially a pharmacokinetic boosting agent, targets the nuclear receptor subfamily 1 group I member 2 (NR1I2), also known as the pregnane X receptor (PXR). The expression of drug-metabolic enzymes and transporters that mediate the responses of mammals to their chemical environment and various endogenous chemicals is regulated by this receptor [63]. Table 1 shows the protein targets of Sartans and the two components of Paxlovid, along with the number of other drugs that bind to the same targets. The full table can be seen in Appendix A.

It should be noted that the proteoforms have nothing to do with drug efficiency, but rather contribute to the reliability of the physical drug-receptor interaction. Additionally, the number of proteoforms is proportional to the number of studies conducted on that specific protein. Consequently, it is no surprise that ACE2 has a great number of proteoforms, as there are an enormous number of studies on that receptor, whereas for the SARS-CoV2 3CLpro, no proteoforms have been identified yet. The full table with all the experimentally supported events for proteoforms can be found in Appendix A.

Based on the Human Protein Atlas, the RNA tissue specificity, protein tissue expression, and subcellular location of the drugs’ protein targets were retrieved (Table 2). Heart, skin, kidneys, blood vessels, skeletal muscles, brain, liver, lungs, and adrenal glands are among the organs where AT1R is expressed [64], whereas c-JUN is mostly overexpressed in cancer tissue. In adipose, spleen, adrenal gland, and the big colon tissue, the highest concentrations of PPAR mRNA have been identified (4–7). Multiple lines of research also suggest that PPARG is crucial for controlling adipocyte differentiation and glucose homeostasis [65]. NR1I2 mRNA concentrations are abundant in the liver, and lower concentrations of it have been found in the small intestine, the colon, the stomach, and skeletal muscles [66]. PXR’s highest levels of expression are encountered in the small intestine, the colon, and the liver [67]. It has been demonstrated that the ACE2 protein is highly expressed in the brush border of enterocytes of the small intestines, but it has also been spotted in lung and endothelial tissue [68,69].

### 3.2. Construction of Protein–Protein and Protein–Drug Interaction Networks

To identify all the proteins that physically interact with the protein targets mentioned above, three Sartans and one Paxlovid were projected into the experimentally supported human interactome (Figure 3a,b). ACE2, the protein that Sartans will target if they are used in the treatment of COVID-19, was also projected into the human interactome in order to identify its first neighbors (Figure 3c). Sartans’ receptors interact with a greater number of proteins, and their interactome consists of 335 nodes and 2989 edges. The main receptor of all Sartans, AT1R, interacts with 48 different proteins, whereas c-JUN (Irbesartan’s receptor) interacts with 186 other proteins and PPAR-γ (Telmisartan’s receptor) with 120 more. The last two receptors also interact with themselves, as they form dimers, and with each other. ACE2, Sartans’ suggested target, and some of its 9 first neighbors form dimers, and its interactome consists of 10 nodes and 18 edges. NR1I2 interacts with itself, thus forming a dimer, as well as with 28 other proteins; thus, its interactome consists of 30 nodes and 126 edges.

Next, the two protein–drug networks for Paxlovid and Sartans that contained the drugs in question, their targets, and their first neighbors, as well as any drugs that bind to them, were constructed (Figure 4 and Figure 5). Sartans’ network consists of 1011 nodes and 4133 edges; its diameter is 6 and its radius is 4. Ritonavir’s network consists of 215 nodes and 254 edges; its diameter is 4 and its radius is 2. In order to better visualize and compare the results, Sartans’ network was divided into three distinct networks: the network of Irbesartan (that binds to AT1R and c-JUN), the network of Telmisartan (that binds to AT1R and PPARG-γ), and the network for all the other Sartans that only bind to AT1R. In the third network, all six Sartans are grouped into one group node. As seen in Figure 6, the six Sartans that only bind to AT1R (i.e., Valsartan, Olmesartan, Candesartan cilexetil, Eprosartan, Losartan, and Azilsartan medoxomil) have 49 first neighbors and 159 drugs possibly interacting with them; their networks’ diameter is 4 and their radius is 2. Irbesartan, that binds to both the AT1R and c-JUN, has 236 first neighbors and 607 drugs possibly interacting with it; the network’s diameter is 6 and its radius is 3. Telmisartan, which binds to both the AT1R and PPARG-γ, has 170 first neighbors and 397 drugs possibly interacting with it; the network’s diameter is 6 and its radius is 3. The interactome of Sartans that exclusively target AT1R was compared with the Protein–Drug Network of Paxlovid. To ensure a valid network comparison, ACE2 and its neighboring proteins, along with drugs that bind to them, were added to the network of Sartans that exclusively target AT1R (Figure 7). The resulting network has 272 nodes and 420 edges; its diameter is 5 and its radius is 3. The two targets combined have 107 interactors, and there are 163 drugs that may interact with the Sartans.

Finally, the protein–drug network of Perphenazine that contained the drug, its three protein targets (i.e., Dopamine Receptor D1 (DRD1), Dopamine Receptor D2 (DRD2), and Calmodulin 1 (CALM1)), their first neighbors, as well as any drugs that bind to them, was constructed (Figure 8). Perphenazine’s network consists of 787 nodes and 1182 edges; its diameter is 6 and its radius is 3. The three protein targets have a total of 151 first neighbors, and there are 633 drugs possibly interacting with Perphenazine. In Appendix A, the synoptic table (closeness centrality, betweenness centrality, and node degree) of the analysis of each protein–drug network can be found.

### 3.3. Shared First Neighbor Distance Metrics (Jaccard Index)

In Table 3, the Jaccard index and Jaccard distance between the shared first neighbors of the two drugs in question and the third drug (Perphenazine) can be seen.

### 3.4. Proteoform Identification of Protein Drug Targets and Proteins Involved in Sartans’ and Paxlovid’s Based Interactomes

The term “proteoform”, established by Smith et al., is used to describe “all of the different molecular forms in which the protein product of a single gene can be found, including changes due to genetic variations, alternatively spliced RNA transcripts and post-translational modifications” [70]. In this work, we focused on the proteoforms that result from PTMs, allelic variants of genes, and alternative splicing of the two drugs’ protein targets and their interactors. The full tables of the experimentally supported events of PTMs of both the drugs’ receptors and their first neighbors can be seen in Appendix A.

In Figure 9, it can be seen that the PTMs of Sartans’ protein targets result from modified residues (excluding lipids, glycans, and protein crosslinks) (37.78%), glycosylation (covalently attached glycan group(s)) (24.44%), disulfide bonds (cysteine residues participating in disulfide bonds) (11.11%), chain (extent of a polypeptide chain in the mature protein) (11.11%), cross-link (residues participating in covalent linkage(s) between proteins) (11.11%), lipidation (covalently attached lipid group(s)) (2.22%), and signal (sequence targeting proteins to the secretory pathway or periplasmic space) (2.22%). The PTMs of Sartans’ protein targets, if repurposed for the treatment of COVID-19, arise from glycosylation (41.67%), disulfide bonds (20.83%), chain (20.83%), modified residues (8.33%), lipidation (4.17%), and signal (4.17%). According to UniProt, the protein target of Paxlovid does not get modified post-translationally.

In Figure 10, the PTMs of the interactors AT1R, ACE2, and NR1I2 are presented. The PTMs of the first neighbors of AT1R are a result of modified residues (59.36%), chain (11.87%), disulfide bonds (10.27%), glycosylation (7.76%), cross-link (3.65%), lipidation (3.20%), initiator methionine (cleaved initiator methionine) (2.74%), signal (0.91%), and transit peptide (extent of a transit peptide for organelle targeting) (0.23%). The PTMs of the first neighbors of ACE2 arise from modified residues (54.95%), chain (8.11%), peptide (extent of an active peptide in the mature protein) (7.21%), glycosylation (7.21%), cross-link (5.41%), disulfide bond (5.41%), signal (4.50%), lipidation (2.70%), initiator methionine (2.70%), and propeptide (part of a protein that is cleaved during maturation or activation) (1.80%). The PTMs of the interactors of NR1I2 are a result of modified residues (76.70%), cross-link (10.92%), chain (7.52%), initiator methionine (cleaved initiator methionine) (2.91%), lipidation (0.97%), signal (0.49%), and glycosylation (0.49%).

The proteoforms arising from allelic variations are 5 for the AT1R receptor gene and 13 for the PPARG-γ receptor gene. The full tables of allelic variations can be seen in Appendix A.

Out of Sartans receptor genes, the AT1R gene has 6 transcript variants that result from alternative splicing, PPARG-γ has 16, ACE2 has 6, and Paxlvovid’s receptor gene has 3. The full tables of alternative transcripts can be seen in Appendix A.

### 3.5. Functional and Disease Annotation of Proteins Involved in Sartans’ and Paxlovids’ Based Interactomes

Functional annotation analysis revealed that AT1R, in addition to its angiotensin type I receptor activity, is associated with acetyltransferase activator activity, bradykinin receptor binding, angiotensin type II receptor activity, peptide receptor activity, G-protein-coupled receptor binding, protein heterodimerization activity, and enzyme activator activity. The TF c-JUN is related to R-SMAD binding, Rho and Ras GTPase activator activity, RNA polymerase II transcription factor activity, enhancer binding, double-stranded DNA binding, promoter binding, DNA regulatory region binding, structure-specific DNA binding, and transcription coactivator activity, amongst others. Finally, PPAR-γ is linked with arachidonic acid, eicosatetraenoic acid, and eicosanoid binding, transcription activator binding, prostaglandin, and retinoid X receptor binding, prostanoid receptor activity, retinoic acid receptor binding, fatty acid binding, and more. The molecular function Gene Ontology (GO) terms that are associated with Paxlovid’s human receptor, NR1I2, are steroid hormone receptor activity, ligand-dependent nuclear receptor activity, and drug binding. Finally, ACE2 is associated with glycoprotein binding as well as viral receptor, carboxypeptidase, and exopeptidase activity. The full table with all the over-represented GO terms of the drugs’ protein targets, along with the GO-IDs and *p*-values, can be seen in Appendix A.

As far as the first neighbors are concerned, the ten molecular function GO terms associated with Sartans’ and Paxlovid’s receptors that had the best *p*-values were chosen. As it can be seen below, AT1R’s, c-JUN’s, and PPAR-γ’s first neighbors are related to transcription factor binding, protein domain specific binding, acetyltransferase activity, phosphothreonine residue binding, transferase activity, catalytic activity, acting on DNA, peptide butyryltransferase activity, peptide crotonyltransferase activity, and histone H2B acetyltransferase activity. NR1I2′s first neighbors are linked with transcription factor binding, transcription regulator activity, nuclear steroid receptor activity, DNA binding, nuclear thyroid hormone receptor binding, nuclear estrogen receptor binding, transcription coregulator binding, acetyltransferase activity, DNA-binding transcription factor activity, and STAT family protein binding. If Sartans are repurposed against COVID-19, they are going to have two protein receptors, the AT1R and the ACE2, because only the sartans that originally had one protein target were chosen. The two receptors’ first neighbors are related to exogenous protein binding, G protein-coupled receptor binding, adenylate cyclase regulator activity, phosphatase activator activity, protein-containing complex binding, beta-adrenergic receptor kinase activity, titin binding, dopamine receptor binding, molecular function regulator activity, and binding (Table 4). The full table with all the over-represented GO terms of the first neighbors of the drugs’ protein targets, along with the GO-IDs, *p*-values, fold enrichment, and FDR, can be seen in Appendix A.

The gene–disease association of the protein targets of the drugs in question is shown in Figure 11. The AT1R gene (target of Sartans) is mostly associated with neoplasm metastasis, patent ductus arteriosus, aortic aneurysm/abdominal, proteinuria, and hypertensive disease. The JUN gene (target of Sartans) shows a strong association with malignant tumors of the colon, lung, liver, and stomach, as well as osteosarcoma. The PPARG-γ gene (target of Sartans) shows a significant correlation with diabetes mellitus, inflammation, familial partial lipodystrophy type 2, malignant colon tumors, and diabetic nephropathy. The ACE2 gene (target of Sartans if they are repurposed for the treatment of COVID-19) is associated with congestive heart failure, heart failure, myocardial infarction, tubal abortion, and early pregnancy loss. The NRI2 gene (target of Paxlovid) is linked with diabetes mellitus, steatohepatitis, drug-induced liver disease, adenocarcinoma, and obesity.

The gene–disease association of the interactors of the protein targets of the drugs in question is shown in Figure 12. The first neighbors of the targets of AT1R, cJUN, and PPARG-γ are mainly related with malignant neoplasms of the breast, polycythemia vera, Von Hippel–Lindau Syndrome, primary myelofibrosis, and Burkitt lymphoma. The first neighbors of AT1R (Sartans protein target) show a strong association with mast syndrome, leopard syndrome, essential thrombocythemia, uveal melanoma, and acute myelocytic leukemia. The first neighbors of ACE2 (Sartans protein target if they are repurposed for the treatment of COVID-19) are mostly associated with breast carcinoma, melanoma, malignant neoplasms of the breast, hypertensive disease, and ventricular tachycardia. The first neighbors of NR1I2 (target of Paxlovid) are linked with colorectal carcinoma, obesity, Fanconi renotubular syndrome with maturity-onset diabetes of the young, thrombocytopenia, and cardiomyopathy. The full table with all the diseases associated with every target and its first neighbors ranked by disease score can be found in Appendix A.

### 3.6. Connectivity of AT1R, ACE2, and NR1I’s Interactors

The calculation of the node degree of the first neighbors of Sartans’ two receptors, AT1R and ACE2 (when repurposed for COVID-19), as well as Paxlovid’s receptor (NR1I2), was performed. In Table 5, the ten nodes with the highest node degree as well as the mean node degree of all the first neighbors can be seen. The mean node degree for AT1R’s first neighbors is 74, and the mean node degree for ACE2’s first neighbors is 170. NR1I2’s first neighbors mean node degree is 153. The full table with the network analysis (closeness centrality, betweenness centrality, and degree) of the protein–drug networks can be seen in Appendix A.

## 4. Discussion

Sartans are antihypertensive drugs that act on the RAAS by targeting the AT1R and activating the AT2R [10]. Apart from lowering blood pressure, ARBs are associated with lung endothelial protection and inflammation reduction [11]. Lately, in silico studies for drug repurposing for the treatment of COVID-19 have been made, as they constitute a fast and cost-effective plan of action [12,13]. Sartans, as well as the recently developed biSartans, antagonize the AT1R, and therefore it can be assumed that they will also bind to ACE2, the protein that interacts with the S-protein of SARS-CoV-2 and is the entry point of the virus into the cell [12,13]. Pfizer’s Paxlovid, a combination of Nirmatrelvir and Ritonavir, is currently approved by the FDA for emergency use in the treatment of mild to moderate cases of COVID-19 in certain adult and pediatric patients. Paxlovid is, however, a potent CYP3A4 inhibitor, and because this isoenzyme is responsible for the metabolism of many common drugs that are prescribed daily, drug interactions with this antiviral drug could occur [16,17,18].

In the past few years, efforts have been made to repurpose Sartans and a new generation of angiotensin receptor-blocking drugs with a Sartan scaffold called biSartans. Nevertheless, before the process of the development of novel antiviral agents is initiated, it is of utmost importance to carefully consider the potential for off-target interactions (i.e., interactions occurring between the drug and unintended protein targets), as they are a major contributor to drug toxicity. Examples reported regarding drug off-target interactions include inhibition of cytochrome P450 enzymes, alteration of protein kinase activity, interference with ion channels, disruption of transporter function, and alteration of hormone signaling [71]. The identification of these off-target interactions is essential for the establishment of drug safety and the minimization of unintended consequences, such as preclinical and clinical toxic events. Hitherto, no in silico study has been performed to predict the toxicity risks of Sartans or a novel class of synthetic antihypertensive drugs referred to as biSartans in the case that they target ACE2 and are repurposed to treat COVID-19. In this work, a network-based bioinformatics approach was applied to investigate the potential of known FDA-approved Sartans for protein off-target interactions, their unwanted involvement in various biological processes, and their potential interactions with other drugs. It has been proposed that the first neighbors of a drug’s protein target in the human interactome could act as off-targets either for that specific drug or for other drugs targeting the same protein or proteins at close proximity in the human interactome. Consequently, these drugs have a high probability of participating in a drug–drug interaction event or sharing the same side effects [25,72,73]. A comparative analysis of the protein–drug associations for both Sartans and Paxlovid was performed to ascertain the potential for off-target interactions and any possible interactions with other drugs. Additionally, the different proteoforms of the drugs’ protein targets and their first neighbors were studied to investigate the possibility of their interference with drug binding. Furthermore, the functional and disease annotation of both drugs’ protein targets and their first neighbors was performed so that their undesired participation in a variety of biological processes and their possible implication in disease onset could be pinpointed.

On the one hand, Appendix A shows that all three of Sartans’ targets are also targeted by 32 additional drugs. Moreover, if Sartans were to be repurposed for COVID-19 treatment, they would target ACE2, which is already the target of 4 drugs. On the other hand, Paxlovid, which only targets NR1I2, is bound by 52 other drugs. The latter further validates the fact that Paxlovid may cause various drug interactions with common drugs, as they share a receptor (i.e., NR1I2). The protein-drug interaction networks demonstrate that Sartans’ drug targets have 319 first neighbors and 682 drugs that could interact with them or their first neighbors, while Paxlovid has 185 drugs that could interact with it, and its target is directly interacting with 28 more proteins. More specifically, the six Sartans that only bind to the AT1R (i.e., Valsartan, Olmesartan, Candesartan cilexetil, Eprosartan, Losartan, and Azilsartan medoxomil) have 49 first neighbors and 159 drugs possibly interacting with them. Irbesartan, which binds to both the AT1R and c-JUN, has 236 first neighbors and 607 drugs possibly interacting with it. Telmisartan, which binds to both the AT1R and PPARG-γ, has 170 first neighbors and 397 drugs possibly interacting with it. The interactome of the six Sartans that exclusively target AT1R was compared to Paxlovid’s network. For their proper comparison, as mentioned above, ACE2 was added as a protein target, and in the new network, the two protein targets together have 107 first neighbors and 163 drugs that could target them or their first neighbors. From that, it can be seen that even if Sartans target ACE2, the number of drugs that could potentially interact with them through the PPI network is almost the same. Additionally, it can also be observed that the six Sartans that only interact with the AT1R have fewer drugs that could possibly interact with them than Paxlovid, suggesting that if they are indeed used as a treatment for COVID-19, they may have fewer side effects due to drug-drug interactions. Nevertheless, because Sartans are also repurposed for the treatment of diabetic nephropathy, ischemic stroke, ventricular dysfunction, and more, their potential for off-target interactions should be considered before they are repurposed outside the scope of the original medical indication.

From the Jaccard index calculation, it can be seen that the network of Sartans, when used as anti-COVID-19 drugs compared to the network of Paxlovid have the smallest similarity in terms of shared first neighbors. The same thing is observed in the comparison of Paxlovid and Perphenazine, which is an anti-psychotic drug totally unrelated to ACE2 binding. The network of Sartans when used as antihypertensive drugs and their network when used as an anti-COVID-19 strategy share the most first neighbors, which validates the results of this method. The protein targets of Sartans, when they are used as antihypertensive drugs, mostly undergo post-translational modifications that result in modified residues, glycosylation, and cross-links. These modified residues are mainly phosphoserine, phosphothreonine, and N6-acetyllysine, whereas the glycans are attached to the N-terminal end of asparagine and the oxygen of threonine. The cross-links on their end result in glycyl lysine isopeptide formation. On the contrary, the most common PTMs of the receptors of Sartans, if they are repurposed for the treatment of COVID-19, are glycosylation, disulfide bond formation, and differences in chain lengths. In this case, the addition of glycans leads to N-linked (GlcNAc) asparagine formation, and disulfide bonds are formed between cysteines in the extracellular region of ACE2. ACE2 can be found in two forms: the cellular, which is membrane-bound, and the circulating, soluble one. For the formation of the soluble form, the full-length ACE2 is processed either by the metalloprotease ADAM Metallopeptidase Domain 12 (ADAM12) or the Transmembrane Serine Protease 2 (TMPRSS2), resulting in two slightly different circulating forms that lack a small fragment of the C-terminus [74]. The allelic variations of the AT1R gene that result in different receptor isoforms are associated with hypertension and renal tubular dysgenesis, while the allelic variations of the PPARG-γ are linked with severe obesity, type 2 diabetes mellitus and insulin resistance, somatic colon cancer, and type 3 partial familial lipodystrophy. When Sartans are used as antihypertensive drugs, two of their targets (i.e., AT1R and PPARG-γ) are alternatively spliced and result in 22 alternatively spliced transcripts, whereas if they are repurposed for the treatment of COVID-19, they result in 12 alternatively spliced transcripts. Paxlovid’s receptor mRNA also undergoes alternative splicing, which leads to the formation of three alternative transcripts.

The interactors of the protein targets of Sartans, when the drugs are used as antihypertensive drugs, mostly undergo post-translational modifications that result in modified residues, disulfide bond formation, and differences in chain lengths. These modified residues are mainly phosphoserine, phosphotyrosine, and phosphothreonine. On the contrary, the most common PTMs of interactors of the ACE2 receptor of Sartans, if they are repurposed for the treatment of COVID-19, are modified residues, glycosylation, disulfide bond formation, and cross-links. These modified residues are mainly phosphoserine, phosphotyrosine, and phosphothreonine or are modified on the N6 atom. In the case of ACE2 interactors, the addition of glycans leads to N-linked (GlcNAc) asparagine formation. The interactors of NR1I2 (i.e., Paxlovid’s receptor) mostly undergo PTMs that result in modified residues, different chain lengths, and an initiator methionine cleavage. These modified residues are, for the most part, phosphoserine, phosphotyrosine, and phosphothreonine, but they are also residues modified on the N6 atoms and asymmetric dimethylarginines. The cross-links on their ends result in glycyl lysine isopeptide formation.

There has been some evidence suggesting that phosphorylation could have an impact on drug binding, affinity, and consequently clinical efficacy, and that impact is expected to be stronger the closer the phosphorylation is to the drug binding pocket [75]. Furthermore, acetylation may drastically alter protein function via modification of its properties, such as hydrophobicity, solubility, and surface properties [76]. In the case of N-acetylation, which is one of the most common types, protein-protein interactions could be altered, resulting in a higher binding affinity with substrates, cofactors, and other macromolecules [77]. Until recently, the disulfide bond, where two cysteines spontaneously form a bond under redox conditions, was the most well-known covalent crosslink [78]. These bonds contribute to protein stability and are therefore crucial for protein structure and function, as well as the regulation of protein activities [79]. However, apart from disulfide bonds, in proteins there can be other types of cross-links, such as ester and isopeptide bonds [78]. The isopeptide bonds, which were originally discovered about 10 years ago, are peptide bonds that are developed outside the main protein chain [80]. They are intramolecular bonds that form autocatalytically between the side chains of lysine and asparagine/aspartic acid under hydrophobic conditions and provide great stability and structural firmness [78]. Finally, proteins are frequently stabilized by protein glycosylation, which extends their half-life and protects them from denaturation or proteolytic degradation [81]. According to some studies, the level of glycosylation is proportional to the stability increase and reduced flexibility of the protein [82] and is also involved in cell membrane formation and cell–cell adhesion [83]. Specifically, *N*-glycosylation affects protein folding, decreases protein dynamics, and most likely leads to increased protein stability [84], whereas *O*-glycosylation may alter protein–protein interactions [85]. Proteoforms can change protein dynamics, protein binding, and metabolism, and this may have an impact on how well drugs work; therefore, they can impair the effectiveness of commonly used medications or render cells resistant to some medications. However, there has not been any experimental data in this situation to suggest that the proteoforms have an effect on the efficacy of either Sartans or Paxlovids. Since it reflects the number of studies that are pertinent to them, the number of proteoforms effectively serves as a verification of the existence of a physical interaction between the medicine in question and its target.

From the gene–disease association of the protein targets of Sartans, when used as antihypertensive drugs, it can be concluded that the major diseases they are associated with are cancer and neoplasm metastasis, diabetes mellitus, and inflammation, whereas if they are repurposed for COVID-19, apart from cancer, they are also linked with heart failure and related complications, as well as pregnancy loss. Paxlovid’s target appears to be involved in the development of diabetes mellitus, liver disease, and obesity.

The proteins associated with Sartans are mainly involved in protein binding and, at a lower percentage, in nucleic acid and DNA binding. PPIs are indispensable for basic cellular functions in living cells, as they carry out a variety of tasks, including altering the kinetics of enzymes, catalyzing metabolic processes, activating or inactivating proteins, modifying their specificity, and regulating the concentrations of transporting molecules [49]. On top of that, PPIs form a vital part of the signaling events that impact cell growth and transformation and, because of their dynamic nature, provide the versatility that characterizes the mechanisms involved in pathway regulation [50]. DRBPs, or DNA and RNA-binding proteins, are a large class of molecules that make up a sizable portion of cellular proteins and play crucial roles in cells [51]. They create complex dynamic multilevel networks that control nucleotide metabolism, gene expression [52], transcription and translation, DNA repair, splicing, apoptosis, and mediate stress responses [51], among other cellular processes. The interactors of Sartans targets, when the drugs are used for the treatment of hypertension, are mostly associated with breast cancer, leukemia, lymphoma, and obesity. However, if these drugs are repurposed as a treatment strategy for COVID-19, their targets’ interactors are affiliated with breast cancer, leukemia, melanoma, thrombocythemia, hypertensive disease, and heart complications.

On the contrary, the proteins associated with Ritonavir (Paxlovid’s component) mostly take part in transcription regulation and demonstrate TF and transcriptional cofactor (COF) activities. On the one hand, TFs are protein molecules that bind to DNA-regulatory sequences (promoters, enhancers, and silencers), which are typically found in the 5′ upstream region of genes, to modulate the rate of gene transcription and subsequently stimulate or inhibit gene expression and transcription, as well as protein synthesis [53,54]. These proteins are essential for embryogenesis and development and are usually found in larger multiprotein complexes [54]. On the other hand, COFs operate as central effectors of transcription activation and gene expression as they transmit regulatory cues from enhancers to promoters [55]. More specifically, they are scaffold proteins, histone/protein-modifying enzymes, and chromatin remodelers that work in concert to arrange the local chromatin structure and balance the neighboring post-translational modifications, which helps to regulate transcription generally [56]. Paxlovid’s receptor interactors have, for the most part, been linked with diabetes, thrombocythemia, leukemia, and cancer.

From the functional annotation analysis, it can be observed that AT1R and ACE2 first neighbors are mainly involved in protein and enzyme binding, whereas NR1I2 first neighbors are mostly involved in transcription regulation. Consequently, if Sartans are repurposed for COVID-19 treatment, they probably will not be involved in different biological processes apart from calcium ion binding. Additionally, Sartans and Paxlovids are going to be involved in different biological processes, which will probably lead to different side effects due to off-target interactions. Additionally, from the gene–disease association analysis, it can be seen that Sartans’ protein targets are linked with cancer either when they are used as antihypertensive or anti-COVID-19 drugs. If they are used for COVID-19, however, they are also implicated in heart failure and pregnancy loss. Paxlovid’s target is associated with completely different diseases. The first neighbors of Sartans’ targets, either when they are used for the treatment of hypertension or COVID-19, are associated with hypertension, breast cancer, and leukemia. Nonetheless, when they are used as a treatment strategy for COVID-19, they are also linked with some different types of cancer (i.e., melanoma) and thrombocythemia. The first neighbors of NR1I2′s targets are also associated with cancer, leukemia, and thrombocythemia, such as the first neighbors of Sartans’ targets if used as anti-COVID-19 medications, but they are also linked with diabetes. When Sartans are used for different purposes, they are mostly associated with the same diseases but also some different ones. The first neighbors of both drugs’ targets (i.e., Sartans and Paxlovid), when used for the treatment of COVID-19, are for the most part associated with the same diseases. It is evident that both drugs are mainly linked to some type of cancer. This is no surprise because it has been computed that for 15.233 out of the 17.371 human genes, there is at least one paper in PubMed that associated them with some type of cancer [86], thus this finding can be considered purely contingent. Even if there is not a study that connects one particular gene with cancer, it is most likely that there will be one in the future. As far as the other associated diseases are concerned, the gene–disease association could serve as an indication of the implications that that specific drug could have either on the development and progression of that particular disease or on the effect it could have on a patient already suffering from it.

According to the node degree results, the centrality of ACE2 is higher but comparable to the centrality of AT1R, but its neighbors are more central than those of AT1R. According to the analysis, if Sartans are repurposed for the treatment of COVID-19, they might have a higher possibility of side effects that result from off-target interactions than when they are used for the treatment of high blood pressure. This is also consistent with the fact that ARBs (Sartans) were engineered as a substitute for patients who could not endure the adverse events of ACE inhibitors, as mentioned above. As far as Paxlovid is concerned, NR1I2 is less central than the Sartans’ drug targets, but its centrality is still comparable to theirs. The centrality of its first neighbors is also comparable to that of ACE2.

The RAAS has been the prime target for the design of drugs for the treatment of hypertension and cardiovascular diseases. ARBs have been discovered and developed to specifically block the AT1R [87]. Pioneering research on RAAS has resulted in the discovery of the first orally active ARB antagonist, Losartan (DUP753), followed by the development of a series of other potent Sartans for the treatment of hypertension [88]. Since ACE2, which is a part of RAAS and the entry point of SARs-CoV-2 in the cell, Sartans bearing anionic groups, tetrazolate, and carboxylate, were investigated as possible antivirals for the treatment of COVID-19, either by blocking entry or replication of the virus [13,22,23]. Extensive clinical studies have shown that ARBs are beneficial in the treatment of hypertensive patients infected by COVID-19 [89,90]. Other studies investigating the mechanism of triggering disease have shown that an imbalance in RAAS in favor of ANG-II deregulates the system, exaggerates the SARS-CoV-2 specific T-cell response, and thus increases COVID-19 severity and mortality [91,92]. One of the main contributors to morbidity is the release of inflammatory cytokines. Consequently, ARBs could be a promising treatment plan not only for COVID-19 but also for autoimmune diseases. ARBs modulate the cytokine production of T helper (Th)1 and Th17 effector cells by converting pathogenic cytokines to regulatory [93,94]. Modulation of cytokine production has also been reported in other studies [95,96]. The conformation of angiotensin, the principal effector of RAAS, led to the development of non-peptide mimetic ARBs and biSartans (Figure 1 and Figure 13) based on structure–activity relationship (SAR) studies, nuclear magnetic resonance (NMR), fluorescence, and molecular modeling techniques [97,98,99,100,101,102]. In all these studies, the arginine residues play a catalytic role in the basic cleavage sites (R685–S686 and R815–S816) for the S-protein’s cleavage and therefore trigger infection induced by proteases, such as furin and transmembrane serine protease 2 (TMPRSS2) [103,104,105]. Furthermore, arginine mutations on RBD enhance the binding of S-protein with ACE2, increasing transmissibility and infectivity [106]. Based on the above, arginine blockers are also potential therapeutics for treating COVID-19 [13,22]. Proteases such as furin, trypsin, TMPRSS2, and 3CLpro are potential targets for designing novel COVID-19 drugs [107,108]. It is worth mentioning that before the implementation of any of those alternative treatments for COVID-19 or autoimmune disorders, their thorough experimental study is of essence. Data provided by the “COVID-19 Drug Interactions” online database, which is an evidence-based drug-drug interaction resource, indicates that the co-administration of Paxlovid and two Sartans (i.e., Valsartan and Irbesartan) may also need close surveillance. In the case of Valsartan, clinically significant interactions have been reported and, thus, require additional monitoring, such as alteration of drug dosage or timing of administration [109] (data produced on 2 February 2023). Finally, data provided by DrugBank’s “Drug Interaction Checker” indicates that the metabolism of Irbesartan can be decreased when combined with Ritonavir [110,111].

## 5. Conclusions

Sartans are antihypertensive drugs that target the AT1R and activate the AT2R to lower blood pressure. Recently, efforts have been made to repurpose them for the treatment of COVID-19 as a fast and cost-effective alternative to drug discovery. Pfizer’s Paxlovid, a combination of Nirmatrelvir and Ritonavir, is currently approved by the FDA for emergency use in the treatment of mild to moderate COVID-19 in certain adult and child patients, but it is also a potent CYP3A4 inhibitor, which could lead to drug interactions. However, given that off-target interactions (i.e., interactions between the medication and undesired protein targets) are a primary cause of drug toxicity, it is crucial to thoroughly assess their possibility before the process of developing novel antiviral medicines is initiated to ensure the reduction of undesired effects. However, to date, no in silico analysis has been carried out to estimate the potential toxic effects of Sartans or new-generation drugs in this family if they target ACE2 and are used to treat COVID-19. This work used a network-based bioinformatics approach to analyze the potential of known FDA-approved Sartans and Paxlovid for protein off-target interactions, drug–drug interactions, and their potential involvement in numerous biological processes, leveraging experimental support from publicly available proteomic interactome and drug-target data. Network analysis revealed that Sartans’ protein targets (i.e., AT1R, c-JUN, PPAR-γ, and ACE2) are also targets for 36 more drugs, whereas Paxlovid’s only human drug target (i.e., NR1I2) binds 52 other drugs. Additionally, the protein–drug interaction networks demonstrate that Sartans’ drug targets have 319 first neighbors and 682 drugs that could interact with them or their first neighbors, while Paxlovid has 185 drugs that could interact with it, and its target is directly interacting with 28 more proteins. Interestingly, the six Sartans that only bind to the AT1R (i.e., Valsartan, Olmesartan, Candesartan cilexetil, Eprosartan, Losartan, and Azilsartan medoxomil) have 107 first neighbors and 163 drugs possibly interacting with them, suggesting that they may have fewer side effects due to drug–drug interactions than Paxlovid if they are repurposed against COVID-19 and a higher possibility for side effects that result from off-target interactions than when they are used for the treatment of high blood pressure. From the proteoform identification, it is evident that Sartans’ protein targets have far more proteoforms than Paxlovid’s target, either when they are used for the treatment of hypertension or COVID-19. From the study of the most connected interactors of both drugs’ targets, it can be seen that the first neighbors of Paxvlovid’s target have fewer proteoforms than the first neighbors of Sartans’ targets if they are used for the treatment of COVID-19. The study of proteoforms is necessary as they could impact protein binding, dynamics, and metabolism. In this case, however, there has not been any experimental evidence that could indicate an impact of the proteoforms on the drug effectiveness of either Sartans or Paxlovid. The number of proteoforms is essentially a verification of the existence of a physical interaction between the drug in question and its target, as it reflects the number of studies relevant to them.

Furthermore, the functional annotation of both drugs’ protein targets and their first neighbors revealed that the proteins associated with Sartans were mainly involved in protein and enzyme binding, whereas the proteins associated with Paxlovid were mostly involved in transcription regulation and demonstrated TF and cofactor activities, so the two drugs will most likely be involved in different biological processes and will probably lead to different side effects due to off-target interactions. Additionally, the gene–disease association showed that Sartans’ protein targets (for either use) are mainly involved in cancer, heart complications, and pregnancy loss, whereas Paxlovid’s target is linked with obesity, liver disease, and diabetes. The first neighbors of Sartans’ targets (for either use) are for the most part linked with the same diseases (i.e., hypertension, breast cancer, and leukemia). Nonetheless, the interactors of Paxlovid’s target are involved with the same diseases as the interactors of Sartans’ targets if used as a treatment for COVID-19. As a result, the targets of Sartans, whether the drugs are used as antihypertensives or as an anti-COVID-19 strategy, are associated with the same diseases. As far as the two COVID-19 therapies discussed in this work (i.e., Sartans and Paxlovid) are concerned, the interactors of both drugs are mainly linked to the same diseases. Since most of the human genes are linked with cancer in one way or another, the focus should be on the non-cancer associated diseases, as the gene–disease association may provide a hint as to the potential effects that a given drug may have, either on the onset or progression of that disease. Based on the results of this study, no specific problem arises from the combined use of Paxlovid and Sartans.

A limitation of this computational study is that, although it is based on experimental data, it cannot fully reproduce the complexity of drug interactions and potential side effects in vivo. Furthermore, while the study provides valuable insights into the potential for off-target interactions and drug–drug interactions, it does not account for other factors that could affect the efficacy and safety of Sartans and Paxlovid in treating COVID-19, such as patient characteristics and dosage. Future studies should aim to integrate in vitro and in vivo data with network-based bioinformatics approaches to provide a more comprehensive analysis of the potential effects of these drugs on treating COVID-19.

## Figures and Tables

**Figure 2 proteomes-11-00021-f002:**
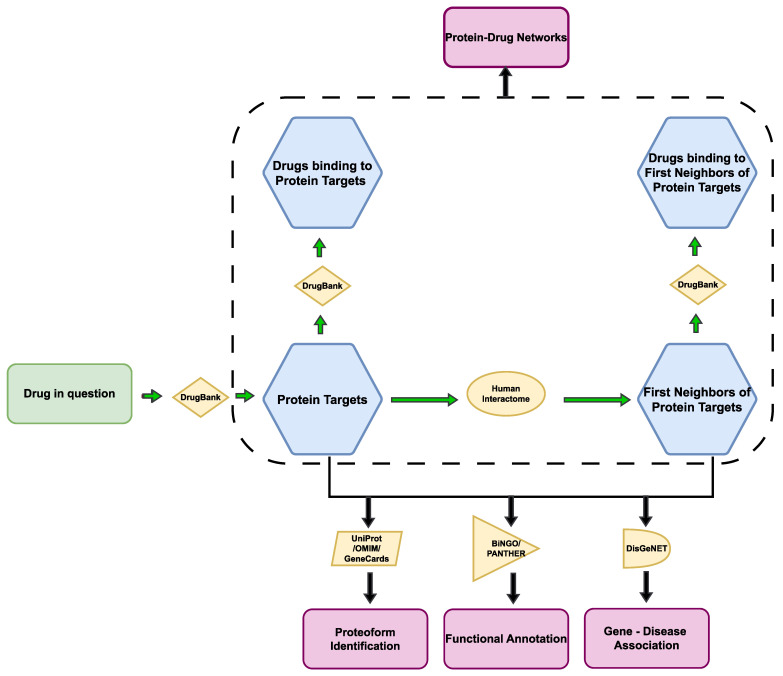
Methodology workflow.

**Figure 3 proteomes-11-00021-f003:**
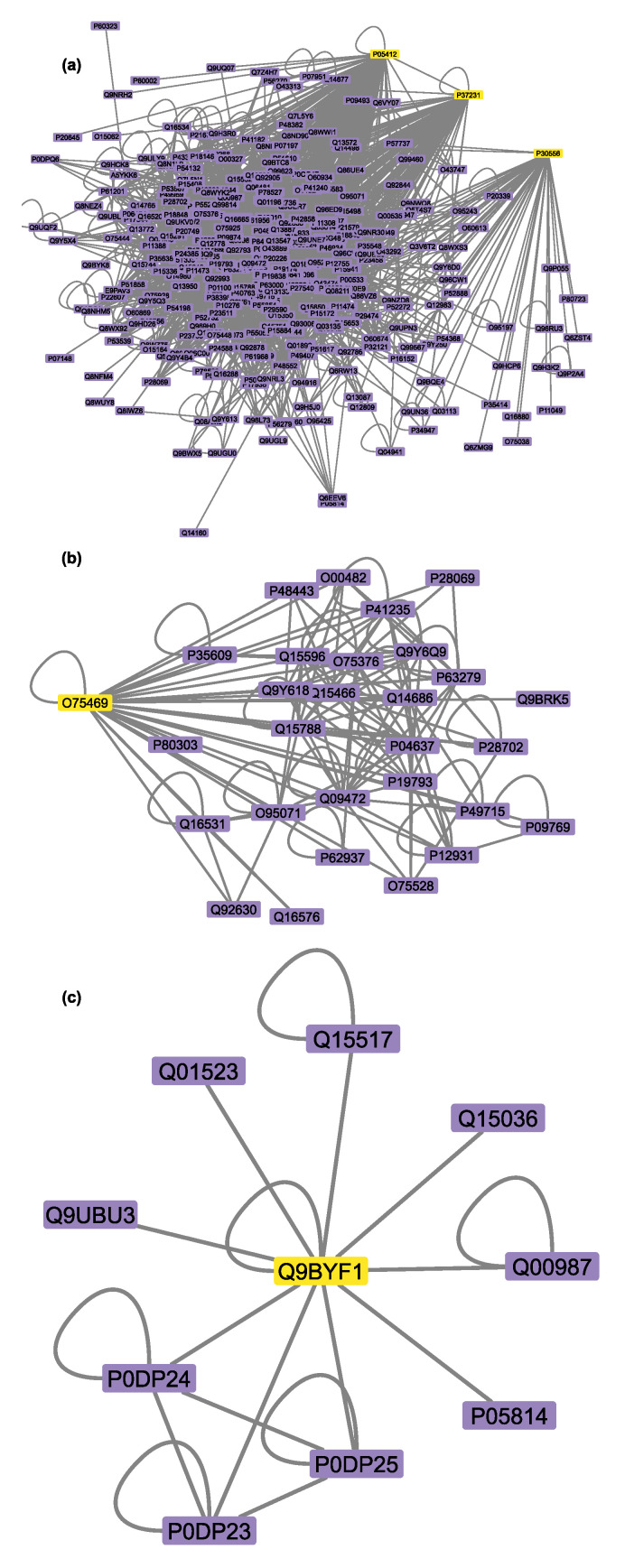
(**a**) PPI network that involves Sartans’ protein receptors (yellow): AT1R (UniProt ID: P30556), c-JUN (UniProt ID: P05412), PPAR-γ (UniProt ID: P37231), and its first neighbors (purple); (**b**) PPI network between Ritonavir’s (Paxlovid’s component) protein receptor: NR1I2 (UniProt ID: 075469) (yellow) and its first neighbors (purple); (**c**) PPI interactome of Sartans’ suggested target when repurposed against COVID-19, ACE2 (UniProtID: Q9BYF1) (yellow).

**Figure 4 proteomes-11-00021-f004:**
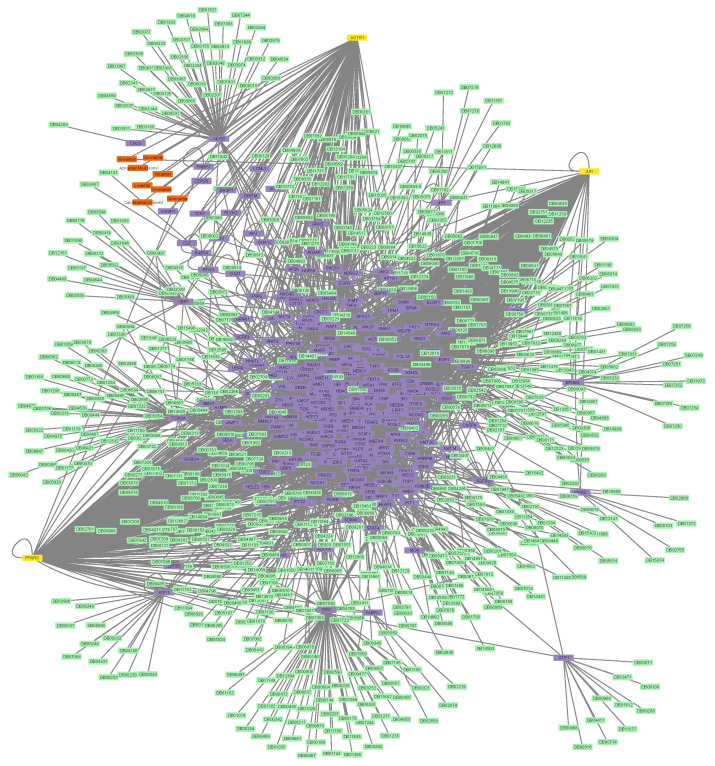
The protein–drug interaction network that involves Sartans (orange), AT1R/c-JUN/PPAR-γ (yellow), the interactors (purple) of Sartans’ targets, and the drugs that target AT1R/c-JUN/PPAR-γ and their interactors (green).

**Figure 5 proteomes-11-00021-f005:**
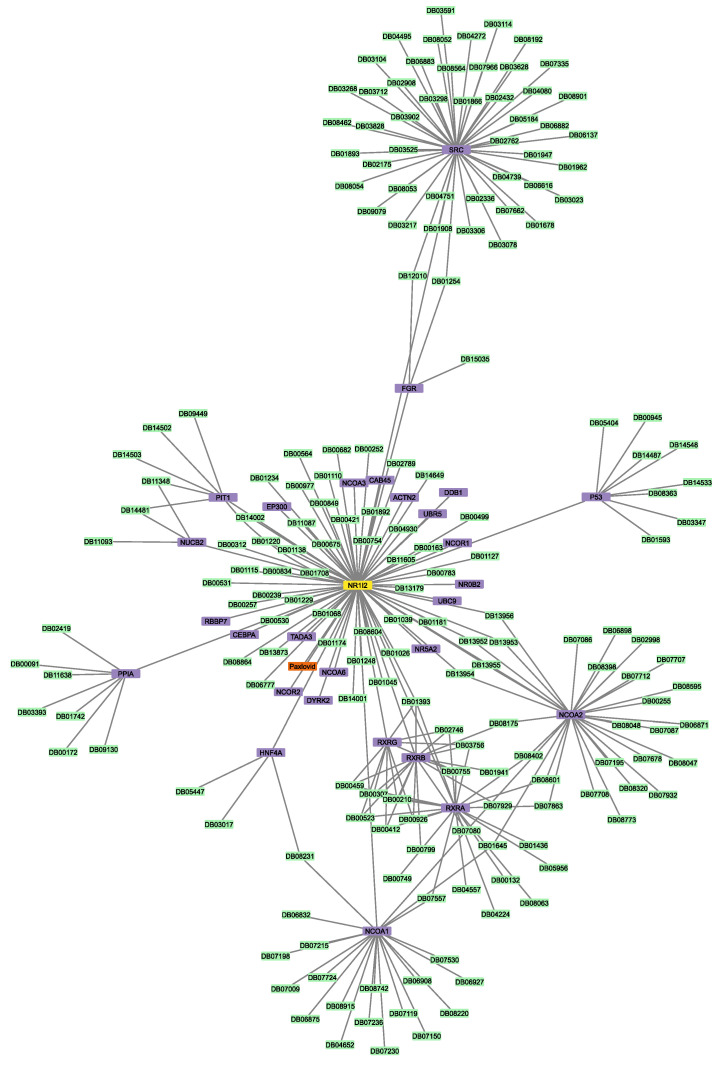
The protein-drug interaction network that involves Ritonavir (Paxlovid’s component) (orange), NR1I2 (yellow), NR1I2′s first neighbors (purple), and the drugs that target NR1I2 and interactors of NR1I2 (green).

**Figure 6 proteomes-11-00021-f006:**
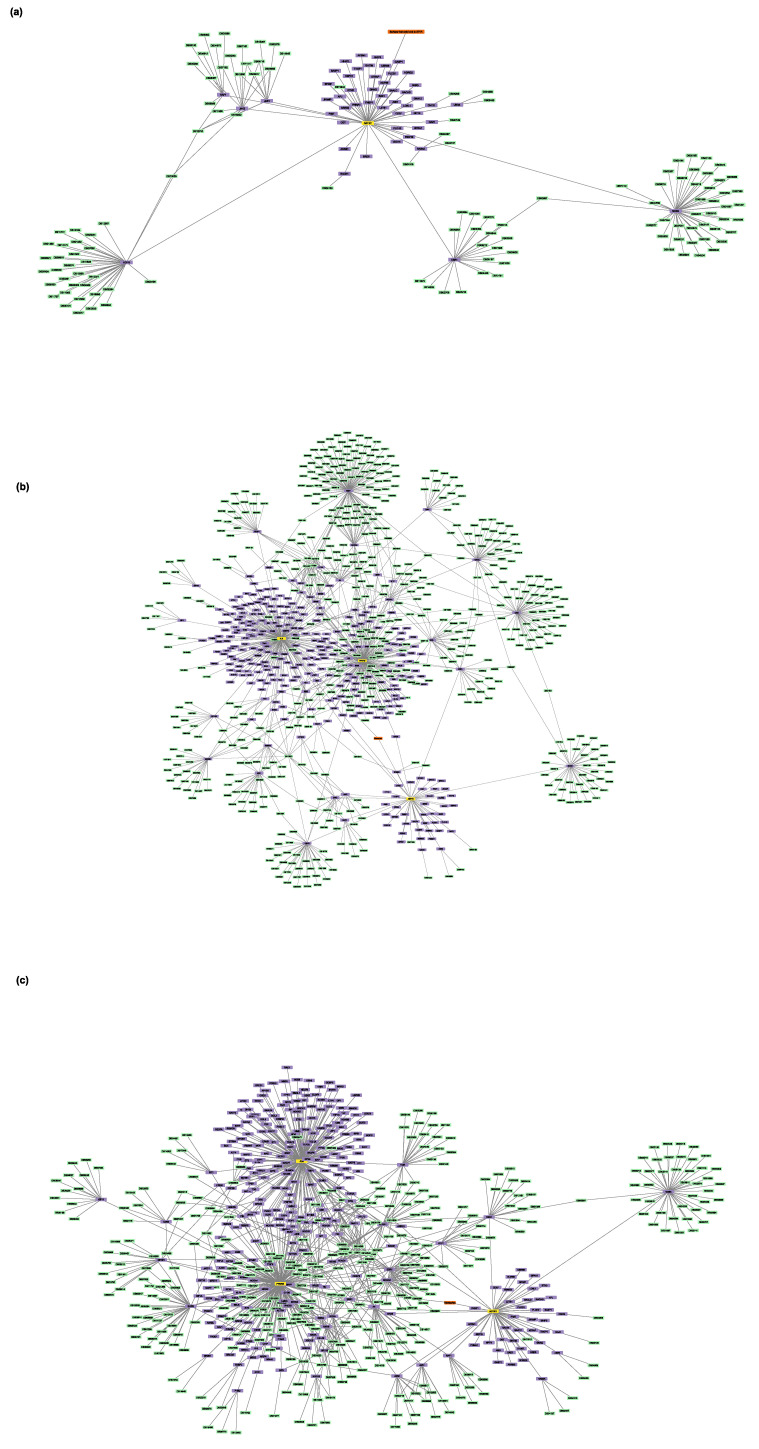
(**a**) The protein–drug interaction network that involves Valsartan, Olmesartan, Candesartan cilexetil, Eprosartan, Losartan, and Azilsartan medoxomil (orange), AT1R (yellow), AT1R’s first neighbors (purple), and the drugs that target AT1R and its interactors (green). Note that all Sartans mentioned above are visualized with one group node: (**b**) the protein–drug interaction network of Irbesartan (orange), AT1R, and c-JUN (yellow), the 2 targets’ first neighbors (purple), and the drugs that target AT1R, c-JUN, and their interactors (green); (**c**) the protein–drug interaction network of Telmisartan (orange), AT1R, and PPARG-γ (yellow), the 2 targets’ first neighbors (purple), and the drugs that target AT1R, PPARG-γ, and their interactors (green).

**Figure 7 proteomes-11-00021-f007:**
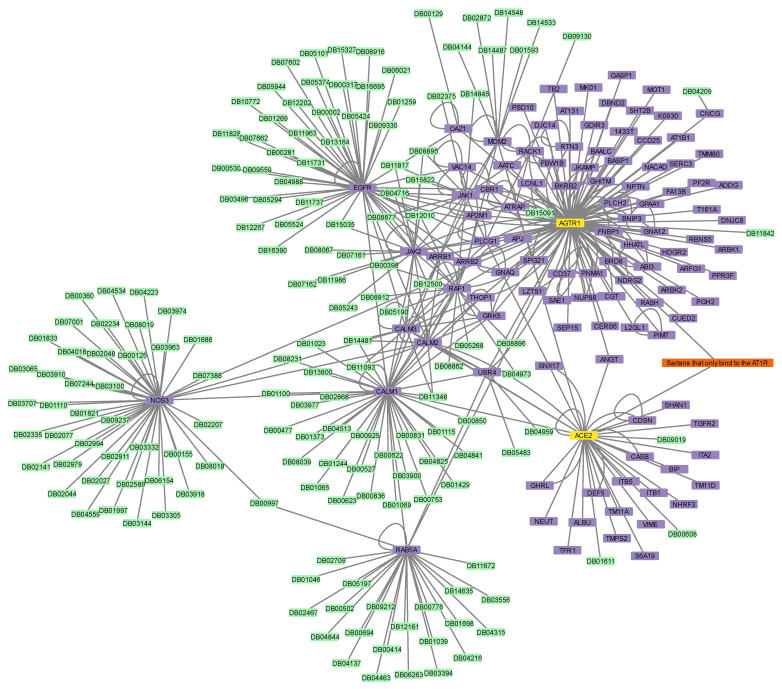
The protein–drug interaction network of Valsartan, Olmesartan, Candesartan cilexetil, Eprosartan, Losartan, and Azilsartan medoxomil (orange), if repurposed for the treatment of COVID-19, their protein targets, AT1R and ACE2 (yellow), AT1R’s and ACE2′s first neighbors (purple), and the drugs that target AT1R and ACE2 and their first interactors (green). Note that all Sartans mentioned above are visualized with one group node.

**Figure 8 proteomes-11-00021-f008:**
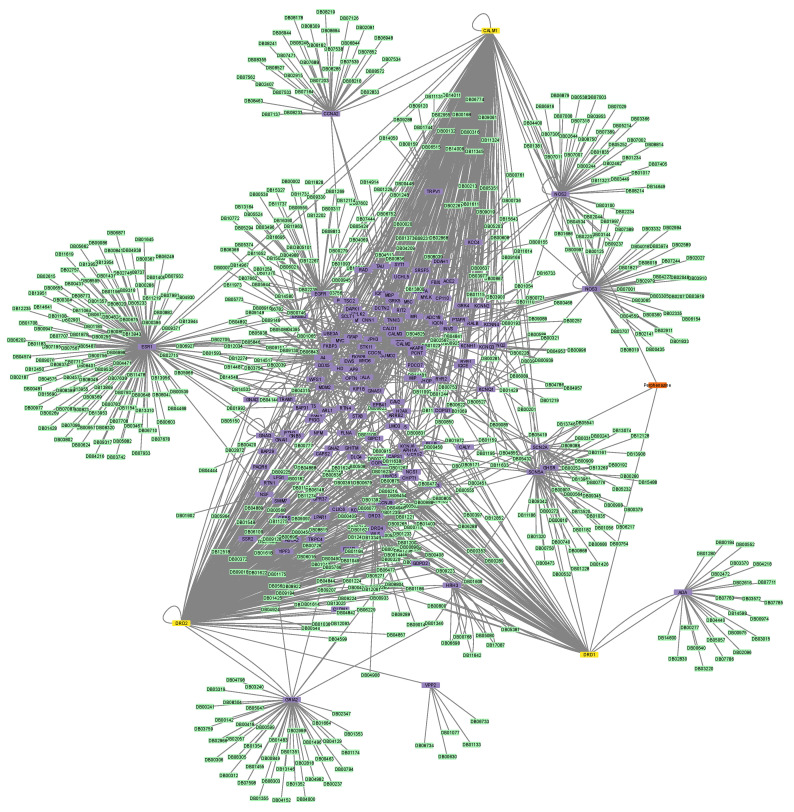
The protein–drug interaction network of Perphenazine (orange), its protein targets, DRD1, DRD2, and CALM1 (yellow), DRD1′s, DRD2′s, and CALM1′s first neighbors (purple), and the drugs that target DRD1, DRD2, and CALM1 and their first interactors (green).

**Figure 9 proteomes-11-00021-f009:**
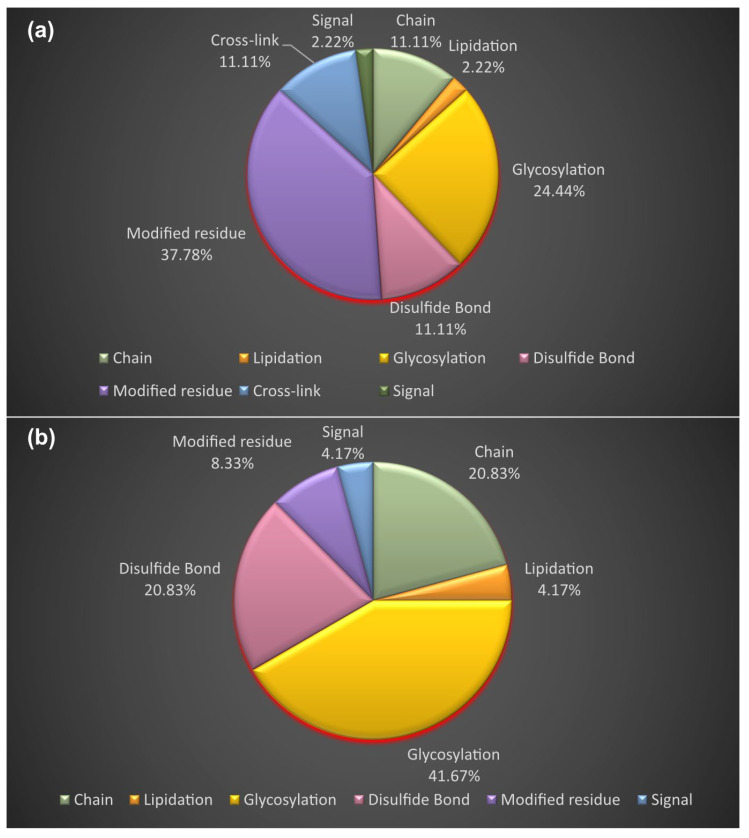
Post-translational modifications of the protein targets of: (**a**) Sartans (AT1R, c-JUN, and PPARG-γ); (**b**) Sartans, if they are repurposed for the treatment of COVID-19 (AT1R and ACE2).

**Figure 10 proteomes-11-00021-f010:**
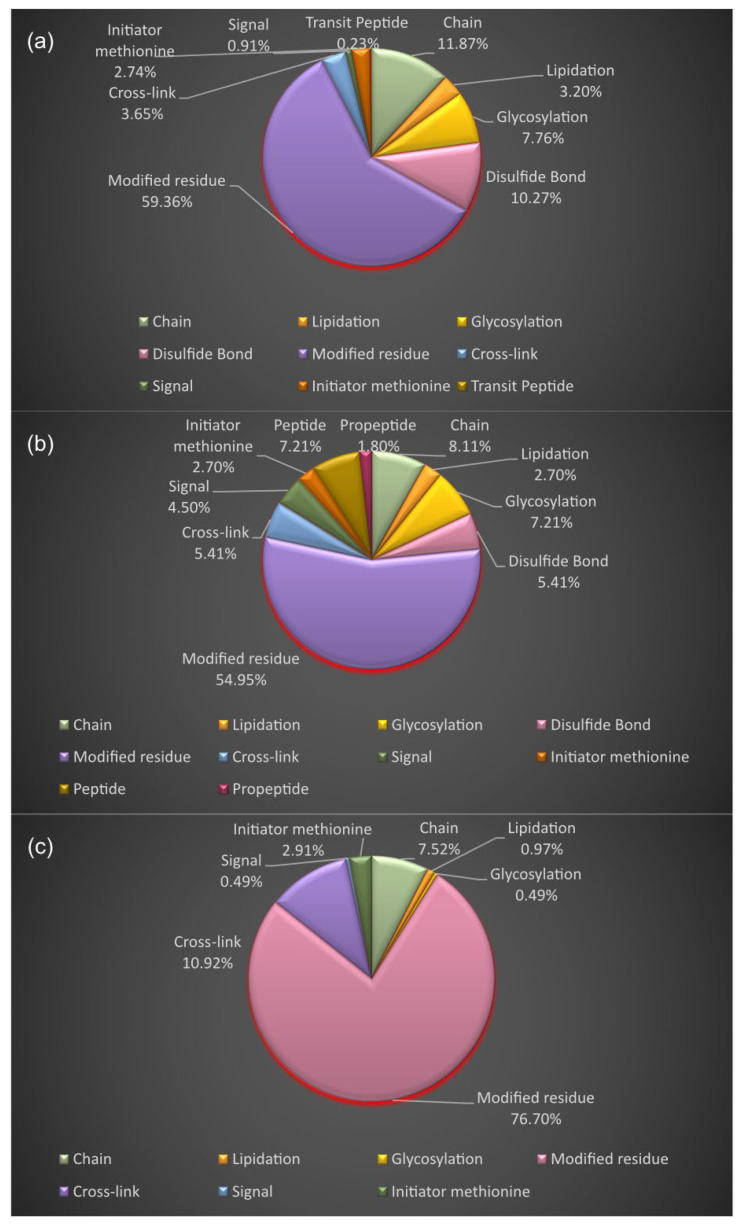
Post-translational modifications of the protein interactors of: (**a**) AT1R; (**b**) ACE2; (**c**) NR1I2.

**Figure 11 proteomes-11-00021-f011:**
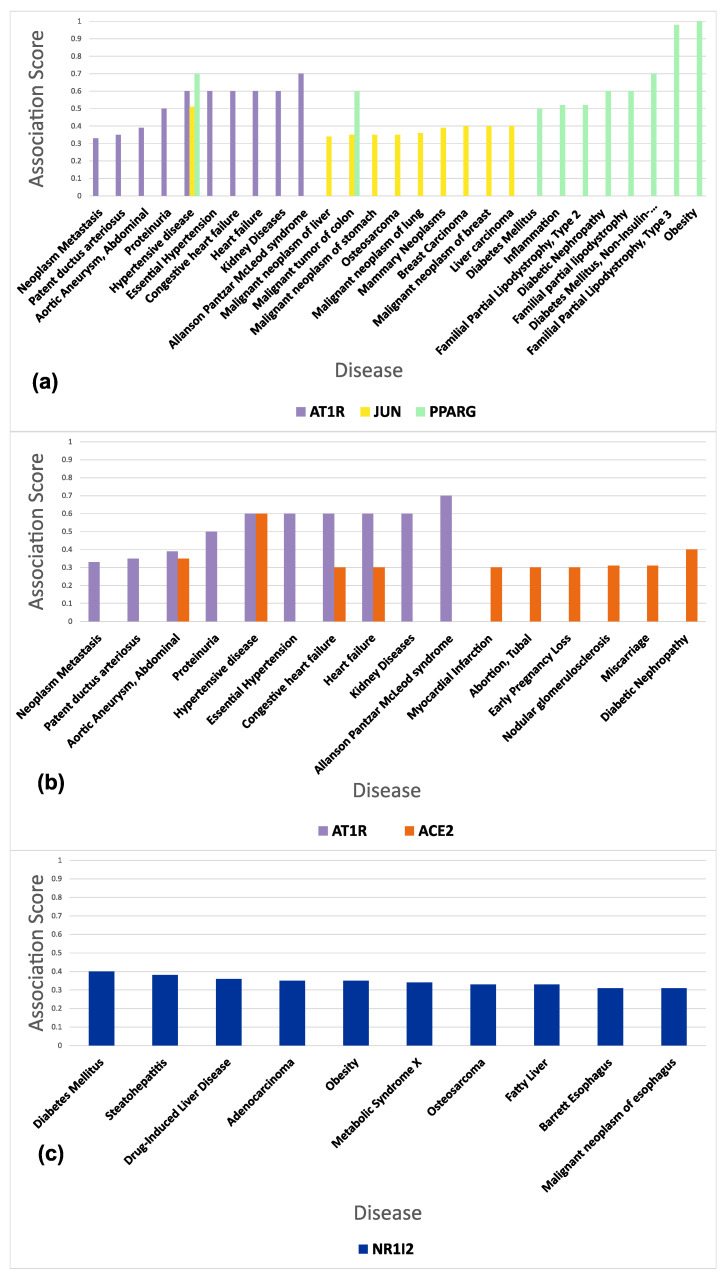
Ten most associated diseases (gene–disease association) of the protein targets of: (**a**) Sartans (AT1R, c-JUN, and PPARG-γ); (**b**) Sartans, if they are repurposed for the treatment of COVID-19 (AT1R and ACE2); (**c**) Paxlovid (NR1I2).

**Figure 12 proteomes-11-00021-f012:**
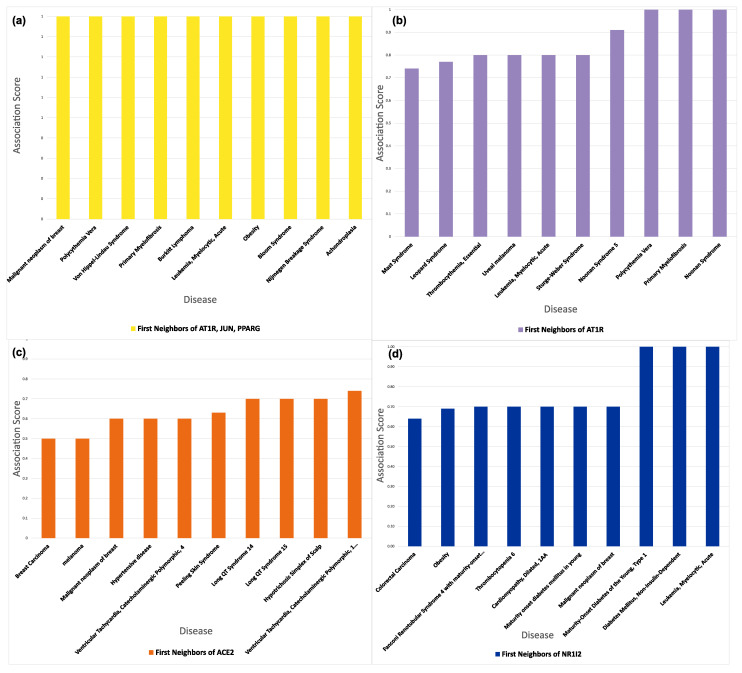
Gene–disease association of the interactors of the protein targets of: (**a**) Sartans (AT1R, c-JUN, and PPARG-γ); (**b**,**c**) Sartans, if they are repurposed for the treatment of COVID-19 (AT1R and ACE2); (**d**) Paxlovid (NR1I2).

**Figure 13 proteomes-11-00021-f013:**
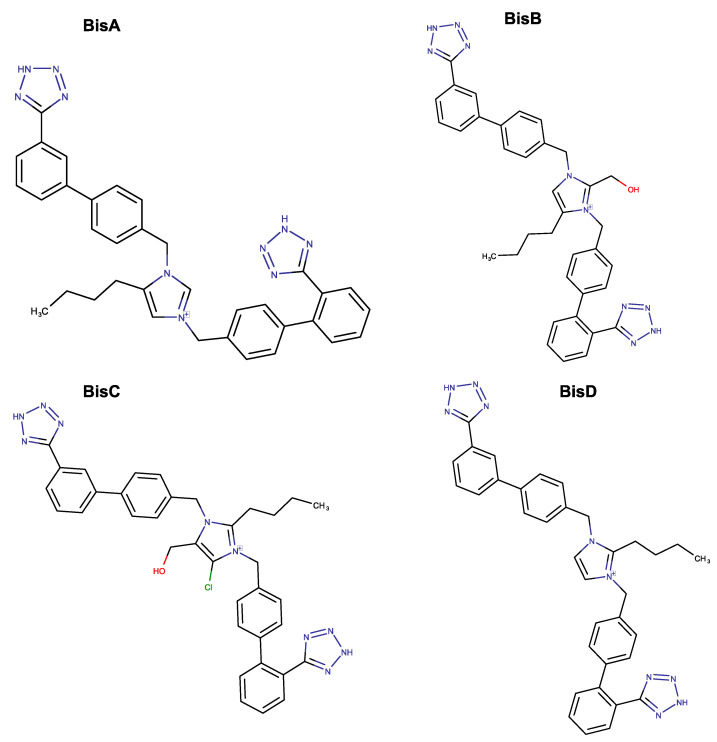
Chemical structures of BisA (4-Butyl-N,N0-bis{[[20-(2H-tetrazol-5-yl)]biphenyl-4-yl] methylimidazolium bromide), BisB (4-Butyl-2-hydroxymethyl-N,N0-bis{[20-(2H-tetrazol-5-yl)-biphenyl-4-yl]- methyimidazolium bromide), BisC (2-Butyl-4-chloro-5-hydroxymethyl-N,N0-bis{[20-(2Htetrazol-5-yl)biphenyl-4-yl]methylpimidazolium bromide) (Dialkylated losartan), BisD (2-Butyl-N,N0-bis{[20-(2H-tetrazol-5-yl)biphenyl-4-yl]methyl}x In imidazolium bromide). Note that the tetrazole group is deprotonated at physiological pH [13,22,23]. Illustrations were made with MarvinSketch, version 22.22.0 [24].

**Table 1 proteomes-11-00021-t001:** Sartans’ and Paxlovid’s (Nirmatrelvir and Ritonavir) receptors, the number of drugs that bind to them, and their number of experimentally supported events of proteoforms. If Sartans target the ACE2 receptor while being repurposed for the treatment of COVID-19, it is shown in bold.

Drug Name	UniProt Entry Name	Organism	Protein Name	Number of Drugs	Experimentally Supported Events of Proteoforms ^c^
Sartans ^a^	AGTR1_HUMAN	HUMAN	Type-1 angiotensin II receptor	9	19
	JUN_HUMAN	HUMAN	Transcription factor AP-1	4	20
	PPARG_HUMAN	HUMAN	Peroxisome proliferator-activated receptor γ	30	32
	**ACE2_HUMAN ^b^**	**HUMAN**	**Angiotensin-converting enzyme 2**	**4**	**21**
Nirmatrelvir	NR1I2_HUMAN	HUMAN	Nuclear receptor subfamily 1 group I member 2	51	4
Ritonavir	R1AB_SARS2	SARS-CoV-2	3CL^pro^	2	-

^a^ Valsartan, Olmesartan, Losartan, Candesartan cilexetil, Eprosartan, Telmisartan, Irbesartan, and Azilsartan medoxomil. ^b^ If Sartans are repurposed against COVID-19 and target ACE2. ^c^ The proteoforms that were taken into consideration result from post-translational modifications, allelic variants, and alternative splicing.

**Table 2 proteomes-11-00021-t002:** Sartans’ targets (when used as anti-hypertensive or anti-COVID-19 drugs) and Paxlovid’s targets are RNA tissue specificity, protein tissue expression, and subcellular location.

Protein Target	RNA Tissue Specificity	Protein Tissue Expression	Subcellular Location
AGTR1	Tissue enhanced (liver, placenta)	Cytoplasmic expression in adipocytes and endothelial cells.	Vesicles (Membrane)
JUN	Low tissue specificity (overexpressed in cancer tissue)	Nuclear expression in several tissues, mostly in a fraction of the cells.	Nucleoplasm (Intracellular)
PPARG	Tissue enhanced (adipose tissue)	Cytoplasmic and nuclear expression in several tissues.	Nucleoplasm, Vesicles (Intracellular)
ACE2	Tissue enhanced (gallbladder, intestine, kidney)	Membranous expression in proximal renal tubules, intestinal tract, seminal vesicle, epididymis, exocrine pancreas, and gallbladder. Expressed in Sertoli and Leydig cells, and trophoblasts. Membranous expression in ciliated cells in nasal mucosa, bronchus, and fallopian tube. Expressed in endothelial cells and pericytes in many tissues.	Membrane, Secreted to blood (different isoforms)
NR1I2	Group enriched (intestine, liver)	Not available	Nucleoplasm (Intracellular)

**Table 3 proteomes-11-00021-t003:** Jaccard index and Jaccard distance between the shared first neighbors’ distribution of Sartans, Paxlovid, and Perphenazine.

DrugA	DrugB	Shared (M11)	Unique to DrugA (M10)	Unique to DrugB (M01)	Jaccard Index (J)	Jaccard Distance (dJ)
Sartans	Paxlovid	17	303	11	0.05136	0.94864
Sartans (COVID-19)	Paxlovid	0	60	28	0	1
Sartans	Perphenazine	19	301	132	0.042035	0.957965
Sartans (COVID-19)	Perphenazine	12	308	139	0.026144	0.973856
Paxlovid	Perphenazine	0	28	151	0	1
Sartans	Sartans (COVID-19)	51	269	9	0.155015	0.844985

**Table 4 proteomes-11-00021-t004:** The 10 over-represented GO terms with the best *p*-value scores for the interactors (first neighbors) of the protein targets of Sartans, Paxlovid, and Sartans when used as an anti-COVID-19 medication.

Drug	Go Term	Fold Enrichment	*p*-Value	FDR
Sartans	transcription factor binding (GO:0008134)	11.23	3.09 × 10^−80^	1.56 × 10^−76^
	binding (GO:0005488)	1.23	3.87 × 10^−26^	7.54 × 10^−24^
	protein domain specific binding (GO:0019904)	4.47	1.08 × 10^−17^	1.65 × 10^−15^
	acetyltransferase activity (GO:0016407)	8.22	2.43 × 10^−8^	1.81 × 10^−6^
	phosphothreonine residue binding (GO:0050816)	60.73	8.13 × 10^−5^	3.77 × 10^−3^
	transferase activity (GO:0016740)	1.70	2.94 × 10^−5^	1.49 × 10^−3^
	catalytic activity, acting on DNA (GO:0140097)	3.31	1.46 × 10^−4^	6.27 × 10^−3^
	peptide butyryltransferase activity (GO:0140065)	60.73	1.54 × 10^−3^	5.01 × 10^−2^
	peptide crotonyltransferase activity (GO:0140064)	60.73	1.54 × 10^−3^	4.98 × 10^−2^
	histone H2B acetyltransferase activity (GO:0044013)	60.73	1.54 × 10^−3^	4.86 × 10^−2^
Paxlovid	transcription factor binding (GO:0008134)	19.75	4.99 × 10^−19^	1.26 × 10^-15^
	transcription regulator activity (GO:0140110)	7.26	8.88 × 10^−13^	8.98 × 10^-10^
	nuclear steroid receptor activity (GO:0003707)	>100	8.43 × 10^−8^	3.55 × 10^−5^
	DNA binding (GO:0003677)	4.22	3.08 × 10^−7^	8.65 × 10^−5^
	nuclear thyroid hormone receptor binding (GO:0046966)	97.93	1.26 × 10^−7^	4.53 × 10^−5^
	nuclear estrogen receptor binding (GO:0030331)	51.95	3.19 × 10^−5^	5.37 × 10^−3^
	transcription coregulator binding (GO:0001221)	24.48	2.32 × 10^−5^	4.34 × 10^−3^
	acetyltransferase activity (GO:0016407)	22.19	3.58 × 10^−4^	3.94 × 10^−2^
	DNA-binding transcription factor activity (GO:0003700)	4.52	2.64 × 10^−4^	3.11 × 10^−2^
	STAT family protein binding (GO:0097677)	>100	1.72 × 10^−4^	2.12 × 10^−2^
Sartans (COVID-19)	exogenous protein binding (GO:0140272)	57.92	1.13 × 10^−9^	5.73 × 10^−6^
	G protein-coupled receptor binding (GO:0001664)	8.74	7.10 × 10^−8^	1.20 × 10^−4^
	adenylate cyclase regulator activity (GO:0010854)	>100	3.28 × 10^−7^	3.32 × 10^−4^
	phosphatase activator activity (GO:0019211)	>100	4.52 × 10^−6^	2.08 × 10^−3^
	protein-containing complex binding (GO:0044877)	5.85	3.31 × 10^−6^	1.68 × 10^−3^
	beta-adrenergic receptor kinase activity (GO:0047696)	>100	2.49 × 10^−6^	2.10 × 10^−3^
	titin binding (GO:0031432)	>100	1.11 × 10^−6^	8.01 × 10^−4^
	dopamine receptor binding (GO:0050780)	41.76	7.79 × 10^−5^	3.03 × 10^−2^
	molecular function regulator activity (GO:0098772)	2.42	6.97 × 10^−5^	2.94 × 10^−2^
	binding (GO:0005488)	1.19	1.09 × 10^−4^	3.92 × 10^−2^

**Table 5 proteomes-11-00021-t005:** The ten most connected interactors of AT1R, ACE2, and NR1I2 with their node degrees and their number of experimentally supported events of proteoforms.

	UniProtID	Protein Name	Degree	Experimentally Supported Events of Proteoforms ^a^
AT1R (Mean Degree: 74)	P00533	Epidermal growth factor receptor	752	73
	P04049	RAF proto-oncogene serine/threonine-protein kinase	226	21
	P19174	1-phosphatidylinositol 4,5-bisphosphate phosphodiesterase gamma-1	206	15
	Q8ND90	Paraneoplastic antigen Ma1	174	1
	P63244	Receptor of activated protein C kinase 1	171	17
	Q9NZD8	Maspardin	164	2
	P49407	Beta-arrestin-1	140	3
	Q6RW13	Type-1 angiotensin II receptor-associated protein	139	6
	P32121	Beta-arrestin-2	138	6
	Q96CW1	AP-2 complex subunit mu	135	3
ACE2 (Mean Degree: 170)	Q00987	E3 ubiquitin-protein ligase Mdm2	513	14
	P0DP24	Calmodulin-2	401	15
	P0DP23	Calmodulin-1	329	15
	P0DP25	Calmodulin-3	328	15
	Q15517	Corneodesmosin	42	3
	Q9BYF1	Angiotensin-converting enzyme 2	28	15
	Q15036	Sorting nexin-17	28	8
	P05814	Beta-casein	22	7
	Q9UBU3	Appetite-regulating hormone	12	10
	Q01523	Defensin alpha 5	4	9
NR1I2 (Mean Degree: 153)	P04637	Cellular tumor antigen p53	857	33
	Q09472	Histone acetyltransferase p300	551	34
	P63279	SUMO-conjugating enzyme UBC9	548	11
	P12931	Proto-oncogene tyrosine-protein kinase Src	492	9
	Q16531	DNA damage-binding protein 1	241	6
	O75376	Nuclear receptor corepressor 1	168	33
	P19793	Retinoic acid receptor RXR-alpha	136	12
	P35609	Alpha-actinin-2	128	2
	O95071	E3 ubiquitin-protein ligase UBR5	126	36
	Q9Y6Q9	Nuclear receptor coactivator 3	120	21

^a^ The proteoforms that were taken into consideration result from post-translational modifications.

## Data Availability

DrugBank is available online on https://go.drugbank.com/ (accessed on 16 January 2023). PICKLE (Protein InteraCtion KnowLedgebasE) is available online on http://www.pickle.gr/ (accessed on 16 January 2023). UniProt (Universal Protein Resource). Available online: https://www.uniprot.org/ (accessed on 16 January 2023). COVID-19 Drug Interactions (Paxlovid with Sartans) is available online on https://www.covid19-druginteractions.org/downloads/interaction_reports.pdf?interaction_ids%5B%5D=20662&interaction_ids%5B%5D=20657&interaction_ids%5B%5D=20644&interaction_ids%5B%5D=20615&interaction_ids%5B%5D=20632&interaction_ids%5B%5D=20625&interaction_ids%5B%5D=20639 (data produced on 2 February 2023). DisGeNET is available online on https://www.disgenet.org/ (accessed on 31 March 2023). OMIM is available online on https://www.omim.org/ (accessed on 31 March 2023). GeneCards is available online on https://www.genecards.org/ (accessed on 31 March 2023). ProteinAnalysis THrough Evolutionary Relationships (PANTHER) is available online on http://www.pantherdb.org/ (accessed on 10 May 2023). ChemMine Tools is available online on https://chemminetools.ucr.edu/ (accessed on 12 May 2023). The Human Protein Atlas is available online on https://www.proteinatlas.org/ (accessed on 12 May 2023).

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
