# Peer review of "Network-Based Prediction of Side Effects of Repurposed Antihypertensive Sartans against COVID-19 via Proteome and Drug-Target Interactomes"

_proteomes, 2023, doi:10.3390/proteomes11020021_

Round 1

Reviewer 1 Report

Authors studied potential off-targets effects and drug interactions of sartans, known antihypertensive drugs, when considered as covid-19 treatment by in-silico analyses based on drug-protein target and protein target-protein interactions extracted from experimental databases. The same methodology was applied to the drug Paxlovid, a known treatment of covid-19 and results were compared to those obtained with sartans. Potential perturbations of drug binding associated to post-translational modifications (PTMs) of drug targets and target interactors were also studied.  

The methodology is interesting but results gave only hypothesis and no new insights into Sartans’use and their real side effects. Analyses are quasi exclusively descriptive and no clear conclusions and validations of the methodology were provided.

1)      There is a significant lack of information in the material and methods section even if some parameters were reported in previous studies (Chasapis et al. 2022). For example, calculation of the association score should be described to get an idea of the relevance of the analyses. Idem for BINGO Parameters.

2)      What about the significance of enrichment in terms of GO: why not apply statistical tests such as hypergeometric tests for enrichment analyzes in terms of GO?

3)      Number of items for each GO terms in figure 10 not presented (idem for fig. 11). It is difficult to compare results when GO terms are not as the same level (eg. protein binding and enzyme binding)

4)      Targets can be specific of different organs, tissues or biological fluids. This is not discussed.

5)      From lines 530 to 535: results are exposed again in the discussion part but there is not discussion of these results. Conclusions?

6)      In the introduction, it is unclear whether this methodology has been applied to other drugs. Is this methodology original? Other publication can be cited with for example: Introduction of Toxicity prediction using target, interactome, and pathway profiles as descriptors. (2023) Toxicol Lett. 2023 Apr 13;381:20-26. doi: 10.1016/j.toxlet.2023.04.005. Online ahead of print.

7)      Is the number of proteoforms of protein targets and first neighbors relevant to evaluate drug efficiency? Certain proteoforms can be minor in abundance or proportion relatively to the parent protein whereas certain can represent a large part of the protein. Moreover, these modifications can also be cumulative.

8)      Links with the same disease are not clear lines 583-587: “When Sartans are used for different purposes, they are mostly associated with the same diseases, but also some different ones. The first neighbors of both drugs’ targets (i.e., Sartans and Paxlovid) when used for the treatment of COVID-19 are for the most part associated with the same diseases.” This is not really convincing. Idem for the interactors of drug targets lines 695-699.

9)      Line 699: “As far as the two COVID-19 therapies discussed in this work (i.e., Sartans and Paxlovid) are concerned, the interactors of both drugs are mainly linked with the same diseases.” And what can be concluded from this?

10)   Line 582 “like the targets first neighbors of Sartan’s targets, if used as anti-COVID medications…”. It is also the case for leukemia for first neighbors of AT1R, JUN and PPARg.

11)   The entry “Malignant neoplasm of breast” is present two times in the figure 12a.

Author Response

I would like to express my sincere appreciation to the reviewers for their insightful comments, with which we fully agree. Their feedback has been instrumental in improving the quality of our work, and we have made every effort to address their suggestions in the revised manuscript.

# Reviewer 1

Authors studied potential off-targets effects and drug interactions of sartans, known antihypertensive drugs, when considered as covid-19 treatment by in-silico analyses based on drug-protein target and protein target-protein interactions extracted from experimental databases. The same methodology was applied to the drug Paxlovid, a known treatment of covid-19 and results were compared to those obtained with sartans. Potential perturbations of drug binding associated to post-translational modifications (PTMs) of drug targets and target interactors were also studied. 

The methodology is interesting but results gave only hypothesis and no new insights into Sartans’use and their real side effects. Analyses are quasi exclusively descriptive and no clear conclusions and validations of the methodology were provided.

 Thank you very much for your thoughtful comments. We agree with your comments, and we have revised the manuscript accordingly. The nature of the analyses is mostly descriptive as the scope of our study is to provide guidelines for the design of future experiments. In any case, we go more in depth into the quantitative comparison between the generated networks by means of a classical metrics for network comparison (Jaccard Index) that allowed us to give a quantitative estimation of the global superposition between different drugs mechanism of action. With an initial indication of the potential for side effects arising either from off-target or drug interactions and the possible effects the drug administration could have in the onset, development and progression of various diseases the implementation of drug repurposing experiments could be further facilitated. In any case, it is of uttermost importance that all these results are also verified experimentally to ensure safety.

1)      There is a significant lack of information in the material and methods section even if some parameters were reported in previous studies (Chasapis et al. 2022). For example, calculation of the association score should be described to get an idea of the relevance of the analyses.Idem for BINGO Parameters.

 Thank you very much for your comments now they are addressed in the edited manuscript.

2)      What about the significance of enrichment in terms of GO: why not apply statistical tests such as hypergeometric tests for enrichment analyzes in terms of GO?

 We agree and it is addressed in the edited manuscript.

3)      Number of items for each GO terms in figure 10 not presented (idem for fig. 11). It is difficult to compare results when GO terms are not as the same level (eg. protein binding and enzyme binding)

 Thank you very much for your comment and we agree. The number of GO items is added in the edited manuscript. As far as the levels of the GO terms are concerned, a few changes were made. The use of BiNGO that uses the term-to-term approach of the hypergeometric distribution was limited to the GO analysis of the drugs’ receptors as a mere verification of the information encountered in the scientific literature about their function. For the Go analysis of the receptors’ first neighbors, PANTHER that uses Fisher’s exact Test with False Discovery Rate (FDR) correction and also categorizes the terms as parent-child was chosen. In the GO terms table (for the first neighbors of the receptors) that was added, the parent terms with the best p-values were chosen.

4)      Targets can be specific of different organs, tissues or biological fluids. This is not discussed.

 Now it is addressed in the edited manuscript.

5)      From lines 530 to 535: results are exposed again in the discussion part but there is not discussion of these results. Conclusions?

We agree with your comment, and changes have now been made to the edited manuscript.

6)      In the introduction, it is unclear whether this methodology has been applied to other drugs. Is this methodology original? Other publication can be cited with for example: Introduction of Toxicity prediction using target, interactome, and pathway profiles as descriptors. (2023) Toxicol Lett. 2023 Apr 13;381:20-26. doi: 10.1016/j.toxlet.2023.04.005. Online ahead of print.

 We understand your concern  and the references were added to the manuscript.

7)      Is the number of proteoforms of protein targets and first neighbors relevant to evaluate drug efficiency? Certain proteoforms can be minor in abundance or proportion relatively to the parent protein whereas certain can represent a large part of the protein. Moreover, these modifications can also be cumulative.

 Thank you very much for your comment. It is addressed in the edited manuscript.

8)  Links with the same disease are not clear lines 583-587: “When Sartans are used for different purposes, they are mostly associated with the same diseases, but also some different ones. The first neighbors of both drugs’ targets (i.e., Sartans and Paxlovid) when used for the treatment of COVID-19 are for the most part associated with the same diseases.” This is not really convincing.

 We agree and appropriate additions were made to the edited manuscript together with a global quantitative comparison of the sets of ‘first neighbours’ distribution similarities by Jaccard index. Moreover, the insertion of a ‘negative control test’ as an anti-psychotic drug helped to put in context our results.

9) Line 699: “As far as the two COVID-19 therapies discussed in this work (i.e., Sartans and Paxlovid) are concerned, the interactors of both drugs are mainly linked with the same diseases.”  

Thank you very much for your comment. It is addressed in the edited manuscript.

10)   Line 582 “like the targets first neighbors of Sartan’s targets, if used as anti-COVID medications…”. It is also the case for leukemia for first neighbors of AT1R, JUN and PPARg.

 Thank you very much for your comment. Leukemia is common for the first neighbors either when the drug is used for COVID or when used for hypertension. The association with melanoma is distinct for the First neighbors of Sartans targets when used as anti-COVID.

11)   The entry “Malignant neoplasm of breast” is present two times in the figure 12a.

Now it is corrected in the edited manuscript.

Reviewer 2 Report

The manuscript entitled "Network-based prediction of Off-Target Effects for repurposed 2 antihypertensive Sartans against COVID-19 via proteome and 3 drug-target interactomes", by Kiouri et al., is an interesting study of potential side-effects evaluation of Sartans and Paxlovid based on combined in-silico analysis of public data of proteomics & proteoforms, interactomics, protein-drugs binding, functional analyses etc.

The methods adopted and their integration is interesting and the results convincing. My main suggestion is to improve the presentation of the draft. In the current version of the manuscript the results section is too focused on the partial lists of outputs that the integrated bioinformatic method generates (number of interacting proteins, percentage of single PTMs, etc.), without proper general comments on each set of results. At the same time, the discussion section should give a "synoptic" vision of the results previously described, trying to summarize the take-home messages resulting from the integration of the performed studies.

Minor points:

1) Some abbreviations are defined in the text at the first appearance,  others are simply listed at the end of the manuscript. I suggest to employ a uniform criterion.

2) Control typing at line 134 ("of of")

English language is OK (only minor typing errors)

Author Response

I would like to express my sincere appreciation to the reviewers for their insightful comments, with which we fully agree. Their feedback has been instrumental in improving the quality of our work, and we have made every effort to address their suggestions in the revised manuscript.

# Reviewer 2

The manuscript entitled "Network-based prediction of Off-Target Effects for repurposed 2 antihypertensive Sartans against COVID-19 via proteome and 3 drug-target interactomes", by Kiouri et al., is an interesting study of potential side-effects evaluation of Sartans and Paxlovid based on combined in-silico analysis of public data of proteomics & proteoforms, interactomics, protein-drugs binding, functional analyses etc.

The methods adopted and their integration is interesting and the results convincing. My main suggestion is to improve the presentation of the draft. In the current version of the manuscript the results section is too focused on the partial lists of outputs that the integrated bioinformatic method generates (number of interacting proteins, percentage of single PTMs, etc.), without proper general comments on each set of results. At the same time, the discussion section should give a "synoptic" vision of the results previously described, trying to summarize the take-home messages resulting from the integration of the performed studies.

Thank you for your insightful views. In the edited manuscript, we tried to better summarize the results in order to make the take-home message clearer.

Minor points:

1) Some abbreviations are defined in the text at the first appearance, others are simply listed at the end of the manuscript. I suggest to employ a uniform criterion.

 Thank you very much for your comment. It is addressed in the edited manuscript.

2) Control typing at line 134 ("of of")

 Now It is corrected.

Round 2

Reviewer 1 Report

I recommend publishing this work after reading the answers given by the authors. The authors provided answers to the questions posed and more information on the software parameters used. Pay attention to the comment: “11) The entry “Malignant neoplasm of breast” is present two times in the figure 12a.” The error is still there.

Author Response

Thank you very much for recommending  that the work can be published. The error has been corrected in the new version